# Less Is More: Fast and Accurate Reasoning with Cross-Head Unified Sparse Attention

**Lijie Yang** [* 1]  **Zhihao Zhang** [* 2]  **Arti Jain** [2]  **Shijie Cao** [3]  **Baihong Yuan** [2]  **Yiwei Chen** [2]
**Zhihao Jia** [2]  **Ravi Netravali** [1]

## Abstract

Large reasoning models achieve strong performance through test-time scaling, but this incurs substantial computational overhead due to long decoding from short prompts. While sparse attention can reduce latency and memory usage, existing methods often degrade reasoning accuracy because selection errors accumulate over long generation horizons, or require costly retraining. We introduce LessIsMore, a training-free sparse attention mechanism for long-horizon reasoning. Our key insight is that token importance in reasoning is global and stable: critical tokens are largely shared across attention heads and remain stable over decoding steps. Guided by this structure, LessIsMore enforces cross-head unified token selection and preserves recent context via a stable recency window, yielding a globally consistent token set that can be reused across layers. Across multiple model families and challenging reasoning benchmarks, LessIsMore matches or improves accuracy while attending to substantially fewer tokens. With kernel-level optimizations, LessIsMore achieves up to $1.6\times$ end-to-end decoding speedup and up to $1.72\times$ faster sparse attention computation, with additional long-context results demonstrating the generality of our approach. Code is available at https://github.com/DerrickYLJ/LessIsMore.

## 1. Introduction

Recent advances in large reasoning models (LRMs) have substantially improved the ability of LLMs to solve com-

plex, multi-step reasoning tasks. State-of-the-art systems such as DeepSeek-R1 (DeepSeek-AI, 2025), Gemini-2.5-Pro (DeepMind, 2025), OpenAI o3 (OpenAI, 2025), and Qwen3 (Team, 2025) achieve strong performance by leveraging test-time scaling, where models generate long chains of intermediate reasoning to improve accuracy (Wei et al., 2023; AoPS, 2025; Rein et al., 2023). While effective, this paradigm fundamentally changes the computational profile of inference: modern reasoning workloads are decode-heavy, requiring tens of thousands of autoregressive decoding steps from relatively short prompts (Research, 2024).

This shift exposes decoding-time attention as a primary performance bottleneck. Under standard full attention, each newly generated token attends to the entire growing key-value (KV) cache, causing memory access and compute costs to scale linearly with generation length (Liu et al., 2025). For example, using the HuggingFace implementation, DeepSeek-R1-Distill-Llama-8B requires over 20 minutes on an NVIDIA RTX A5000 GPU to generate 32,768 tokens for a single AIME problem. As reasoning models increasingly rely on long derivations, improving decoding efficiency without compromising reasoning accuracy has become a central systems challenge.

Sparse attention is a natural candidate for addressing this bottleneck. By restricting attention to a subset of tokens, sparse attention reduces both computation and KV cache access costs (Cai et al., 2025; Gao et al., 2025). Existing approaches fall into two broad classes: eviction-based methods, which permanently discard tokens deemed unimportant (Li et al., 2024; Xiao et al., 2023; Zhang et al., 2023; Adnan et al., 2024; Cai et al., 2025), and selection-based methods, which retain the full KV cache but dynamically select a subset of tokens during attention computation (Yang et al., 2024; Tang et al., 2024; Hao et al., 2025; Liu et al., 2024; Gao et al., 2025; Yuan et al., 2025). While both classes achieve substantial speedups on standard long-context tasks, they exhibit significant accuracy degradation on reasoning workloads (Gao et al., 2025).

This degradation stems from a key mismatch between existing sparse attention assumptions and the structure of reasoning. Prior methods typically optimize token selec-

---

[*]Equal contribution  [1]Princeton University, Princeton, NJ, USA  [2]Carnegie Mellon University, Pittsburgh, PA, USA  [3]Microsoft Research, USA. Correspondence to: Lijie Yang <ly3223@princeton.edu>, Zhihao Zhang <zhihaoz3@cs.cmu.edu>.

*Proceedings of the 43rd International Conference on Machine Learning*, Seoul, South Korea. PMLR 306, 2026. Copyright 2026 by the author(s).

tion locally—per attention head, per layer, or per decoding step—based on approximate attention scores (Yang et al., 2024; Tang et al., 2024). In long-generation reasoning, however, even small selection errors compound across thousands of decoding steps, leading to attention recall degradation, logical inconsistency, and in some cases longer generation traces (Lee & Hockenmaier, 2025). For instance, while TidalDecode preserves accuracy at extreme sparsity on retrieval benchmarks, it suffers from high accuracy drop on AIME-style reasoning tasks even with low sparsity ((Figures 1a and 5)) (Yang et al., 2024). These observations suggest that reasoning workloads demand not just local-optimal sparse attention, but stable and globally consistent token selection.

Motivated by this gap, we study attention patterns in reasoning models to understand which properties are essential for accurate long-horizon decoding. Our analysis reveals two consistent locality structures that persist across models, layers, and decoding steps. First, we observe strong cross-head spatial locality: despite conventional assumptions that attention heads serve highly specialized roles (Xiao et al., 2024; Yang et al., 2024), reasoning models exhibit substantial overlap in token importance rankings across heads within the same layer. Second, we find pronounced temporal recency locality: recently generated tokens receive consistently high attention scores, and the proportion of attention mass allocated to recent context remains stable throughout decoding. Together, these patterns indicate that reasoning relies on a small, shared set of globally salient tokens that evolves slowly over time (Lee & Hockenmaier, 2025).

These observations lead to a central insight: token importance in reasoning is a global property, not a head-local one. Based on this, we introduce LessIsMore, a training-free sparse attention mechanism designed explicitly for reasoning workloads. Rather than maintaining independent token subsets per attention head or frequently re-optimizing selections at every layer, LessIsMore enforces Cross-Head Unified Sparse Attention (CUSA) with cross-head unified token selection, where candidate tokens proposed by individual heads are aggregated into a single global ranking. This unified selection captures globally important tokens while reducing selection variance and overhead. To preserve reasoning coherence across long decoding trajectories, LessIsMore further reserves a fixed fraction of the token budget for a stable recency window, reflecting the observed temporal locality of reasoning.

LessIsMore is architecture-agnostic and can be integrated into existing selection-based sparse attention frameworks. In this work, we instantiate LessIsMore on top of TidalDecode (Yang et al., 2024), using two CUSA layers to amortize selection cost across many sparse attention layers. Importantly, we conduct detailed analysis to validate the design

choices—including index sharing across heads, recency window allocation, and infrequent re-selection—are not heuristic but direct consequences of the observed spatial and temporal locality structures in reasoning attention.

We evaluate LessIsMore on multiple model families, including GQA-based DeepSeek-R1-Distill-Llama-8B (DeepSeek-AI, 2025) and Qwen3 models (4B, 8B, and 14B) (Team, 2025) , across diverse reasoning benchmarks such as AIME-24/25 (AoPS, 2025), GPQA-Diamond (Rein et al., 2023), and MATH500, as well as MHA-based LongChat-7B-v1.5-32k (Li et al., 2023) on long-context benchmarks. Across all settings, LessIsMore consistently matches or improves accuracy compared to full attention while attending to substantially fewer tokens. Notably, LessIsMore achieves up to 87.5% sparsity on AIME-24 with no accuracy loss and avoids the generation length inflation observed in prior sparse attention methods (Gao et al., 2025). With kernel-level optimizations for GQA models, LessIsMore delivers up to $1.6\times$ end-to-end decoding speedup over full attention.

In summary, our contributions are:

- We identify stable spatial and temporal locality structures in reasoning attention, showing that token importance is globally shared across heads and remains stable over decoding steps.

- We propose LessIsMore, a training-free sparse attention mechanism that enforces cross-head unified token selection and stable recency preservation, directly addressing error accumulation in long-horizon reasoning.

- We demonstrate that LessIsMore generalizes across model families and reasoning benchmarks, achieving significant decoding speedups while preserving or improving reasoning accuracy.

**Conflict of Interest Disclosure.** The authors declare no financial conflicts of interest related to this work. While one author is affiliated with Microsoft Research, this affiliation did not influence the design, evaluation, or reporting of the methods presented in this paper.

## 2. Observation: Attention Pattern in Long-Horizon Reasoning

Sparse attention methods rely on the assumption that a small subset of tokens captures most of the attention mass during decoding. While this assumption has been validated for long-context retrieval and language modeling tasks, its applicability to long-horizon reasoning remains poorly understood. In this section, we analyze attention patterns in reasoning models to identify which structural properties are essential for accurate sparse decoding over thousands of generation steps.

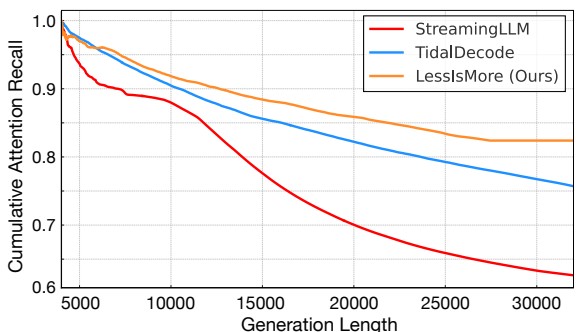

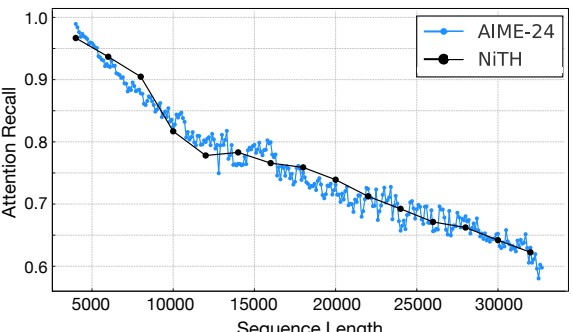

*(a)* Attention recall of different approaches using 4K token budget on an AIME problem.

*(b)* Attention recall of retrieval (NiTH) and reasoning (AIME) tasks.

*Figure 1.* Analysis of attention recall degradation for sparse attention methods on reasoning tasks. Figure 1a compares cumulative average attention recall among StreamingLLM (Xiao et al., 2023), TidalDecode (Yang et al., 2024), and LessIsMore on an AIME-24 reasoning task, using a token budget of 4K and generation length up to 32K tokens on Qwen3-8B. Figure 1b contrasts running-average attention recall of TidalDecode between the reasoning-intensive AIME-24 task throughout the generation and the simpler needle-in-the-haystack retrieval task under the same token budget under various context lengths on Qwen3-8B.

Our analysis reveals that attention behavior in reasoning differs fundamentally from that in standard generation. Rather than exhibiting highly specialized, rapidly changing token importance, reasoning attention is governed by two stable locality structures: cross-head spatial locality and temporal recency locality. These properties persist across models, layers, and decoding steps, and together imply that token importance in reasoning is a global and slowly evolving property. This observation directly challenges the design assumptions of existing sparse attention mechanisms.

### 2.1. Background: Attention Recall as Diagnostic Metric

Attention mechanisms are central to transformer-based LLMs. For each attention head $i$, attention scores and outputs are computed as:

$$W_i = \frac{Q_i K_i^T}{\sqrt{d}}, \quad O_i = \text{softmax}(W_i)V_i, \quad (1)$$

where $Q_i$, $K_i$, and $V_i$ denote the query, key, and value tensors, respectively. Sparse attention methods approximate this computation by selecting a subset of tokens $\rho$ under a fixed budget $k$:

$$\arg\max_{\rho} f(Q_i, K_i[\rho], V_i[\rho], k), \quad |\rho| = k. \quad (2)$$

The effectiveness of sparse attention depends on how well the selected subset captures the true attention distribution. We quantify this using attention recall, defined as the fraction of attention mass retained by the selected tokens:

$$R_i = \frac{\sum(\text{softmax}(W_i)[\rho])}{\sum(\text{softmax}(W_i))}. \quad (3)$$

High attention recall is a necessary condition for preserving model accuracy under sparse decoding, especially when errors can accumulate over long generation horizons.

### 2.2. Failure of Existing Sparse Attention in Reasoning

Existing sparse attention methods typically optimize attention recall locally, either per head, per layer, or per decoding step. While such local approximations are often sufficient for short generation or retrieval tasks, they prove brittle for reasoning workloads that require thousands of consecutive decoding steps.

As shown in Figure 1a, both StreamingLLM and TidalDecode exhibit substantial attention recall degradation on the AIME-24 benchmark, with recall deteriorating steadily as generation length increases. TidalDecode achieves only ~75% recall, while StreamingLLM falls below 65%. Importantly, these failures occur despite using the same token budgets that preserve accuracy on retrieval tasks such as needle-in-the-haystack (Figure 1b).

The key distinction is generation length. Reasoning tasks involve orders of magnitude more decoding steps than retrieval tasks, causing even small selection errors to compound over time. Once critical tokens are omitted, subsequent reasoning steps cannot recover, leading to cascading logical errors and, in some cases, longer generation traces (Gao et al., 2025). These observations indicate that sparse attention for reasoning must prioritize global stability in token selection, not just instantaneous recall.

### 2.3. Spatial Locality Across Attention Heads

A common assumption in sparse attention design is that different attention heads specialize in distinct roles and therefore require independent token subsets (Xiao et al., 2024; Yang et al., 2024). However, our analysis reveals that this assumption does not hold for reasoning workloads.

Section 2.3 visualizes the ground-truth top-4K attended tokens across all 32 attention heads in a decoding step of

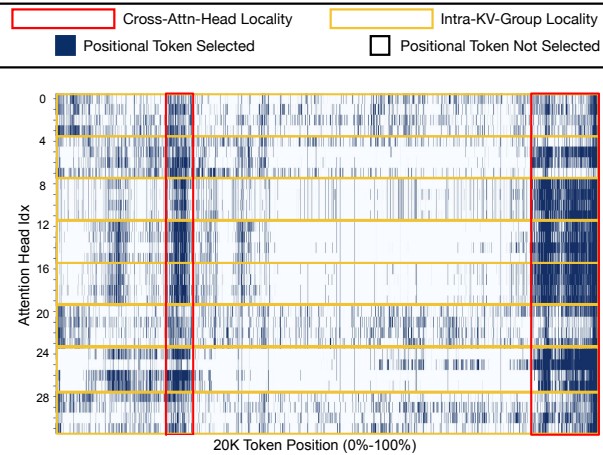

*Figure 2.* The distribution of the ground-truth top-4K tokens across all attention heads at the 20K-th decoding step at Layer 4 on AIME-24 with Qwen3-8B. Dark blue positions indicate tokens included in the ground-truth top-4K budget. We enclose highly overlapped attention regions within the same KV group and across all heads using different colors.

Qwen3-8B. We observe substantial overlap in token importance rankings both within key-value groups (yellow regions) and across all heads (red regions). These overlaps are not transient artifacts: they persist across layers and decoding steps (see Section A.7 and figs. 9 and 10).

This cross-head spatial locality implies that reasoning attention is governed by a shared set of globally important tokens rather than head-specific patterns. Consequently, per-head top-$k$ selection yields only a local optimum, overfitting to head-specific noise while failing to capture globally salient tokens that remain important in future layers. This observation suggests that token selection strategies should aggregate information across heads rather than maintaining independent token subsets.

### 2.4. Temporal Recency Locality

In addition to spatial locality, reasoning attention exhibits a strong temporal structure. As shown in Section 2.3, the most recently generated tokens consistently receive high attention scores across all heads. We further analyze this phenomenon across decoding steps and find that the size of the effective "recency window" remains remarkably stable throughout the reasoning process (Section A.7).

This temporal recency locality reflects the incremental nature of step-by-step reasoning, where the next reasoning step builds directly on the immediately preceding conclusions (Lee & Hockenmaier, 2025). Importantly, our analysis shows that the ratio between the number of recent tokens and the total number of critical tokens remains approximately constant across decoding steps. This property implies that allocating a fixed proportional budget to recent context is important to preserve reasoning coherence as generation length grows.

Prior approaches, such as StreamingLLM, recognize the importance of recent tokens by maintaining a fixed sliding window (Xiao et al., 2023). However, a static window size does not adapt to different token budgets and fails to reflect the stable proportional structure observed in reasoning attention. This motivates selection mechanisms that explicitly reserve a fraction of the token budget for recent context.

## 3. LessIsMore

---
**Algorithm 1** LessIsMore Decoding Pipeline
---
1: **Input:** Current hidden state $h$, KV cache $\mathcal{C}$, token budget $K$, static ratio $r$
2: **Output:** Logits
3: **Initialize:** $\rho = []$
4: **for** each decoder layer $i$ **do**
5: $\quad q, k, v = f(W_{qkv}, h)$
6: $\quad \mathcal{C}.\text{append}(k, v)$
7: $\quad$ **if** $i$ is Full Attention Layer **then**
8: $\quad\quad o = \text{FullAttention}(q, \mathcal{C}[:])$
9: $\quad$ **else if** $i$ is Token Selection Layer **then**
10: $\quad\quad o = \text{FullAttention}(q, \mathcal{C}[:]), P := q \cdot \mathcal{C}.K^\top$
11: $\quad\quad \rho_{\text{head}} = \text{TopKIndices}(P[:, : -(K \cdot r)], k = K \cdot (1-r))$
12: $\quad\quad \rho_{\text{unified}}, \rho_{\text{recent}} = \text{UnionFlatten}(\rho_{\text{head}}), \text{Recent}(K \cdot r)$
13: $\quad\quad \rho = \rho_{\text{unified}}[: K \cdot (1-r)] \cup \rho_{\text{recent}}$
14: $\quad$ **else**
15: $\quad\quad o = \text{SparseAttention}(q, \mathcal{C}[\rho])$
16: $\quad$ **end if**
17: $\quad h = \text{FFN}(o)$
18: **end for**
19:
20: **return** lm_head($h$)
---

Guided by the analysis in Section 2, LessIsMore is designed around a single unifying principle: in long-horizon reasoning, token importance is global across attention heads and remains stable over decoding steps. This principle directly yields two design requirements for sparse attention in reasoning workloads: (i) token selection must be globally consistent across heads, and (ii) token selection must be stable across layers, with explicit preservation of recent context.

LessIsMore practices these requirements through Cross-Head Unified Sparse Attention (CUSA), a training-free sparse attention mechanism that performs cross-head unified token selection and stable recency preservation. While the underlying design is general and can be incorporated into any selection-based sparse attention framework, in this work we instantiate LessIsMore on top of TidalDecode (Yang et al., 2024), a strong baseline that separates token selection from sparse attention computation. LessIsMore uses a small number of token selection layers to compute globally valid token indices, which are then reused

by subsequent sparse attention layers. This reuse directly reflects the observed stability of token importance in reasoning and enables substantial amortization of the selection cost. The complete decoding pipeline is summarized in Algorithm 1 and visually illustrated in Figure 3.

### 3.1. Cross-Head Unified Sparse Attention (CUSA)

As shown in Algorithm 1, LessIsMore processes each decoding step using three types of layers: full attention layers, token selection layers, and sparse attention layers. Full attention layers (lines 7–8) are used sparingly to ensure accurate early-context modeling. Token selection layers (lines 9–15) compute a globally consistent token index set $\rho$, which is then reused by subsequent sparse attention layers (line 16).

At a token selection layer, LessIsMore first computes full attention to obtain the query–key product $P = q \cdot \mathcal{C}.K^\top$ (line 10). Importantly, this computation is used only to estimate token importance, not just to produce attention outputs. The total token budget $K$ is decomposed into two complementary components: (i) a set of globally selected historical tokens and (ii) a reserved set of recent tokens. This decomposition directly mirrors the spatial and temporal locality structures identified in Section 2.

Once computed, the selected token indices $\rho$ are shared by all attention heads and reused across multiple layers. This index sharing is a deliberate design choice: by enforcing a single globally consistent token set, LessIsMore prevents the accumulation of head-specific selection errors and stabilizes sparse attention over long decoding horizons.

### 3.2. Cross-Head Unified Token Selection

To address the strong cross-head spatial locality observed from attention pattern in reasoning (Section 2.3), LessIsMore replaces per-head independent token selection with aggregation through cross-head unified selection.

As implemented in lines 11–12 of Algorithm 1, each attention head independently proposes its top-$k$ candidate tokens based on exact attention scores:

$$\rho_{\text{head}} = \text{TopKIndices}(P[:, : -(K \cdot r)], k = K \cdot (1 - r)).$$

Rather than treating these proposals as disjoint head-specific selections, LessIsMore aggregates them into a single unified candidate set using UnionFlatten($\cdot$). The aggregated tokens are then globally ranked, and only the highest-ranked $K \cdot (1 - r)$ tokens are retained.

This aggregation step enforces a global notion of token importance that reflects agreement across attention heads. Compared to per-head selection, CUSA reduces selection variance, improves attention recall under infrequent reselection, and significantly simplifies KV cache access by allowing all heads to attend to the same token subset. Im-

portantly, CUSA does not assume identical head behavior; instead, it exploits the empirically observed overlap in token importance to identify tokens that are robustly useful for reasoning across heads and future layers.

### 3.3. Stable Recency Preservation

In addition to spatial locality, reasoning attention exhibits strong temporal recency locality (Section 2.4), where recently generated tokens consistently receive high attention across heads and decoding steps. To preserve this structure, LessIsMore explicitly reserves a fixed fraction $r$ of the token budget for recent context.

As shown in lines 12–13 of Algorithm 1, LessIsMore selects the most recent $K \cdot r$ tokens via Recent($K \cdot r$) and unions them with the globally selected historical tokens:

$$\rho = \rho_{\text{unified}}[: K \cdot (1 - r)] \cup \rho_{\text{recent}}.$$

Unlike prior approaches that use a fixed-size sliding window regardless of token budget (Yuan et al., 2025), LessIsMore employs a stable proportional recency window. This design is directly motivated by our observation that the ratio between recent tokens and total critical tokens remains approximately constant throughout reasoning. By allocating a fixed proportion of resources to the recent context, LessIsMore preserves step-by-step reasoning coherence while maintaining sparsity across different budgets and sequence lengths.

### 3.4. Why Infrequent Re-Selection Works

A key advantage of LessIsMore is that token selection does not need to be performed at every decoding layer. Because CUSA produces globally consistent and temporally stable token indices, the selected set $\rho$ remains valid across multiple subsequent layers. This allows LessIsMore to amortize selection cost and reduce overhead without recall drop.

We empirically validate this property in Figure 4, which shows locally optimized selection strategies degrade rapidly when selection is performed infrequently, while LessIsMore maintains high attention recall even when selection is applied only once early in Layer 2 during decoding. This robustness confirms that enforcing global consistency is essential for stable sparse attention in long-horizon reasoning.

## 4. Experiments

### 4.1. Experiment Setup

We conduct extensive experiments to evaluate the accuracy and efficiency of LessIsMore. Our experiments consider four widely-used reasoning models from two families, Qwen3-4, Qwen3-8B, and Qwen3-14B (Team, 2025) and DeepSeek-R1-Distill-Llama-8B (DeepSeek-AI, 2025) backed up with GQA. All models are specifically trained for

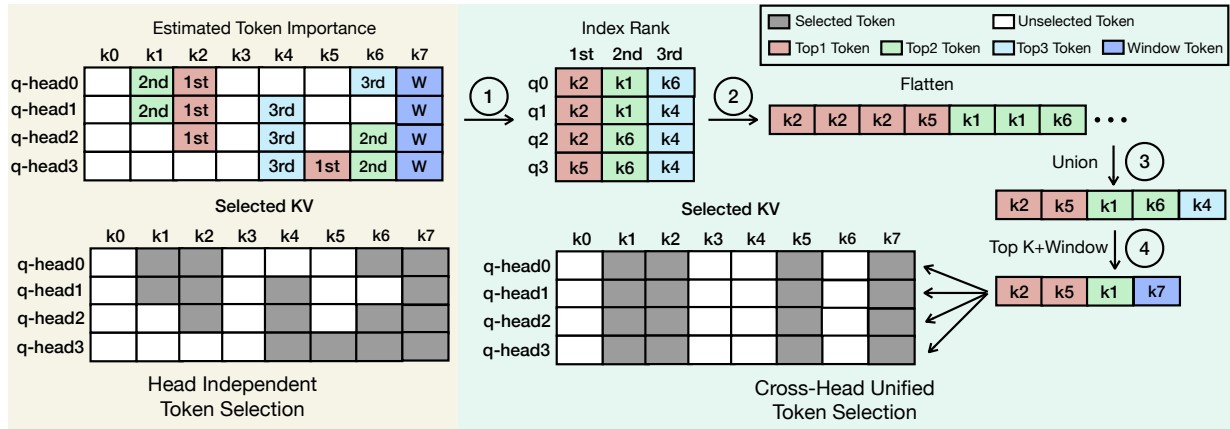

*Figure 3.* Overview of CUSA in LessIsMore. Each attention head first selects its top-$k$ tokens under budget $K = 4$ with $r = 0.25$ reserved for the recency window (Stable Recency Window), producing head-level selections $\rho_{head}$. These indices are then aggregated across heads through Cross-Head Selection into a unified ranked set, from which the top entries are kept. Finally, this unified set is concatenated with the most recent tokens to form the final token set $\rho$, which is shared by all sparse attention layers. Only tokens in $\rho$ are loaded from the KV cache until the next selection step, ensuring both cross-head consistency and stable recency preservation.

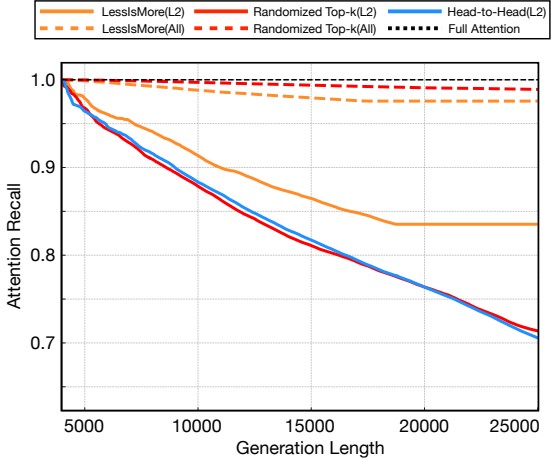

*Figure 4.* The Top-4K (with 1K window size) attention recall of different selection schemes applied only on Layer 2 (L2) or all decoding layers (All). (1) **LessIsMore**: unified top-$k$ selection across all attention heads with 25% tokens allocated to the recency window; (2) **Randomized Top-$k$**: random application of one query head's top-$k$ tokens to the entire KV group; and (3) **Head-to-Head**: direct utilization of top-$k$ tokens for each individual attention head.

reasoning tasks and perform the effective thinking process by generating extensive tokens. Further, we evaluate on multiple mainstream reasoning tasks, including AIME-24, AIME-25, GPQA-Diamond, and MATH500.

**Baselines.** In Section 4.2, we compare LessIsMore against both training-free sparse attention methods (TidalDecode (Yang et al., 2024) and Quest (Tang et al., 2024)) and training-required, reasoning-focused baselines (SeerAttention-r (Gao et al., 2025)). For all training-free approaches, including LessIsMore, we use two initial full-attention layers, followed by a single token selection layer performing head-specific top-$k$ selection for TidalDecode

and a single CUSA layer for LessIsMore.

To ensure a fair and principled comparison, we follow the layer-selection procedure of (Yang et al., 2024) and apply the same one re-selection layer to both LessIsMore and TidalDecode across all benchmarks, balancing accuracy and efficiency. Specifically, we use Layer 12 for DeepSeek-R1-8B, Qwen3-8B and -14B, and Layer 20 for Qwen3-4B. A detailed analysis of this choice is provided in Section A.6.

In all experiments, LessIsMore uses a static recency-window ratio of $r = 0.25$, with an ablation study on the effect of $r$ reported in Section A.1.2. We additionally reserve four tokens as attention sinks in all experiments. For Quest, we follow the original configuration with hybrid attention layers and block sizes of 16 on DeepSeek-R1-8B and 32 on the Qwen model family, applying sparse attention at all layers. For SeerAttention-r, we adopt the settings from (Gao et al., 2025), using a block size of 64 and sparse attention applied at every layer.

**Benchmarks.** We evaluate all methods on challenging reasoning benchmarks, including AIME-24/25, MATH500, and GPQA-Diamond. All models are evaluated with identical prompts and a maximum generation length of 32K tokens to ensure consistent comparison. To reduce evaluation variance, we generate 64 complete traces per problem for AIME-24/25, 8 for MATH500, and 16 for GPQA-Diamond, and report Pass@1 accuracy over all traces. These sample sizes are calibrated to keep evaluation noise under $\pm 0.6\%$ (Section A.2).

In Section 4.4, we compare decoding efficiency for LessIsMore implemented with customized kernels for the selection-based models (Quest, TidalDecode) and full attention with FlashInfer (Ye et al., 2024).

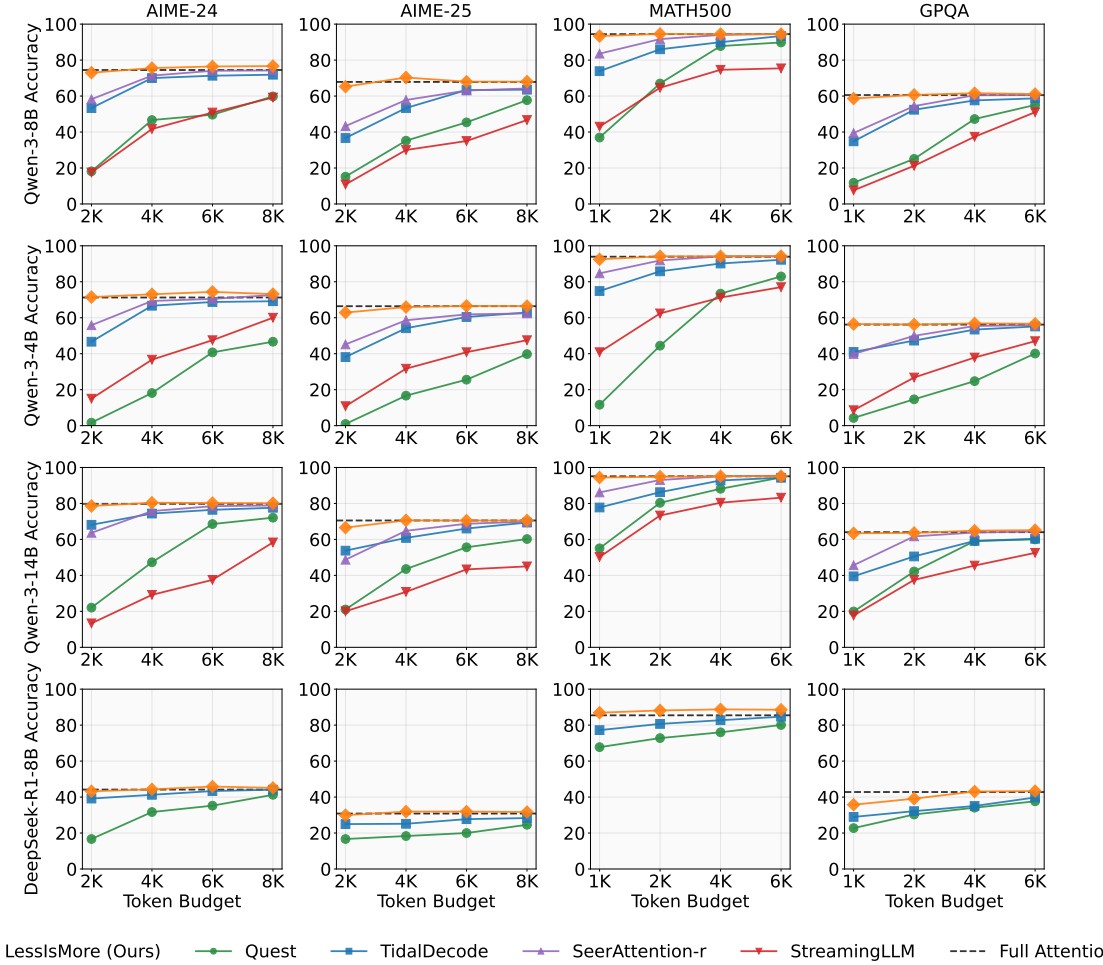

*Figure 5.* Accuracy results of LessIsMore(ours), Quest, StreamingLLM, TidalDecode, SeerAttention-r, and Full Attention across for multiple main-stream reasoning tasks. LessIsMore consistently achieves the lossless accuracy with small token budgets (1K or 2K), always outperforming all others.

## 4.2. Math and Science Reasoning

Figure 5 presents the accuracy comparison among Full Attention, LessIsMore, and other sparse attention methods as baselines across mainstream reasoning benchmarks, AIME-24, AIME-25, MATH500, and GPQA-Diamond. The first three are from the complex mathematical contests, and GPQA-Diamond comprises graduate-level STEM proofing problems. All datasets are evaluated on the reasoning-focused language models DeepSeek-R1-8B and Qwen3-4B, -8B, and -14B. For challenging AIME-24 and -25 tasks, experiments span token budgets of 2K, 4K, 6K, and 8K where the Full Attention model solves problems with an average reasoning length of 15K and 17K tokens, respectively. In contrast, existing sparse attention methods not only suffer accuracy degradation but also extend generation lengths significantly, often requiring 15K–30K tokens to complete the same problems. A similar trend holds for GPQA (Full Attention 8K vs. 8K–18K under baselines) and MATH500 (Full-Attention 5K vs. 5K–16K un-

der baselines). By comparison, LessIsMore consistently achieves the highest accuracy across all evaluated tasks and token budgets, while maintaining generation lengths nearly identical to Full Attention (15K for AIME, 8K for GPQA, 5K for MATH). Specifically, for Qwen3-8B on AIME-24 at the smallest budget (2K tokens), LessIsMore attains nearly lossless accuracy, surpassing Quest, TidalDecode, and training-required SeerAttention-r, all of which suffer notable degradation and longer reasoning traces (Table 3). This dual advantage—accuracy preservation without length inflation—underscores LessIsMore's ability to retain critical contextual information and facilitate fast, accurate reasoning with limited token budgets.

## 4.3. Coding Evaluation

To evaluate generality beyond STEM reasoning, we additionally test LessIsMore on HumanEval (Chen et al., 2021), a code-generation benchmark that requires long-horizon reasoning to produce correct programs. We use the same hyper-

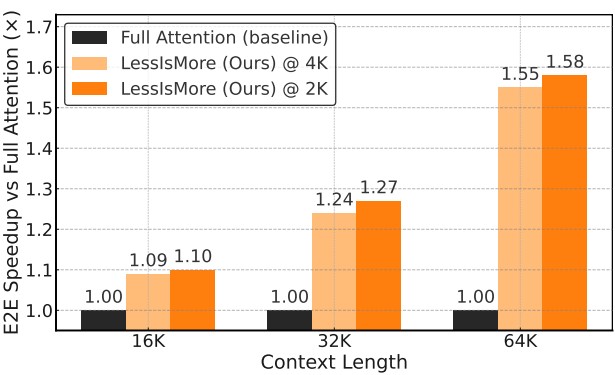
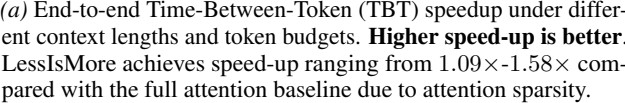
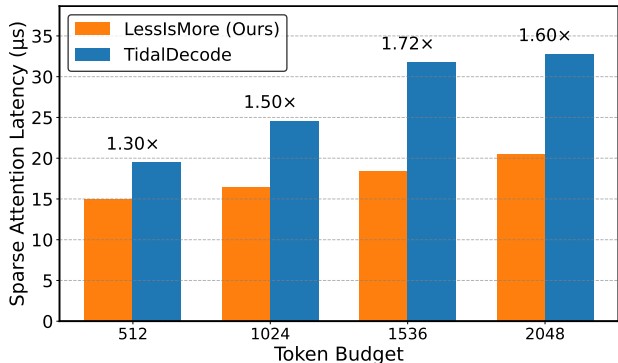

*(a)* End-to-end Time-Between-Token (TBT) speedup under different context lengths and token budgets. **Higher speed-up is better**. LessIsMore achieves speed-up ranging from $1.09\times$-$1.58\times$ compared with the full attention baseline due to attention sparsity.

*(b)* Sparse attention kernel latency comparison between LessIsMore and TidalDecode under different token budgets. **Lower latency is better**, and LessisMore achieves a speed-up ranging from $1.3\times$-$1.72\times$ consistently across all token budgets.

*Figure 6.* Efficiency results with DeepSeek-R1-Distill-LLama-8B on one 80GB NVIDIA A100 GPU.

parameter setup as in Section 4.2 (re-selection layer $= 12$, recency ratio $r = 0.25$) and report Pass@1 accuracy on Qwen3-8B over 4 runs across all 164 problems. As shown in Table 1, LessIsMore attains lossless accuracy at a 2K token budget and matches Full Attention across all evaluated budgets, while TidalDecode degrades by up to $3\%$ under the same constraints. This result indicates that the spatial and recency locality structures we exploit are general properties of long-horizon reasoning workloads, including code generation.

*Table 1.* HumanEval Pass@1 accuracy (164 problems, 4 runs) on Qwen3-8B.

| Method | 2K | 4K | 6K |
|---|---|---|---|
| Full Attention | 88.41 | 88.41 | 88.41 |
| TidalDecode | 85.36 | 87.19 | 88.41 |
| LessIsMore (Ours) | **89.63** | **88.41** | **89.63** |

### 4.4. Efficiency Evaluation

To evaluate the practical efficiency gains of LessIsMore, we implement customized kernels on top of the state-of-the-art attention kernel library FlashInfer (Ye et al., 2024) for GQA-based models. We conduct end-to-end time-between-token (TBT) and kernel-level latency analysis under practical serving setups and report the results in Figure 6. Our evaluation uses DeepSeek-R1-Distill-LLama-8B (AI, 2024b; DeepSeek-AI, 2025) on one 80GB NVIDIA A100 GPU.

For end-to-end speed-up gains, as shown in Figure 6a, compared to the Full Attention baseline, LessIsMore consistently accelerates decoding, delivering **1.1×**, **1.3×**, and **1.6×** end-to-end speedups at 16K, 32K, and 64K context lengths, respectively. These improvements highlight LessIsMore's ability to preserve reasoning quality while providing meaningful computational savings. Given that modern

reasoning models—both open- and closed-source—already support generation lengths well beyond 100K tokens (OpenAI, 2025b; Anthropic, 2025; OpenAI, 2025a), the efficiency advantages of LessIsMore are expected to scale even further in real-world deployments. For kernel-level speed-up gains, as shown in Figure 6b, we observe that LessIsMore can consistently achieve speed-ups from **1.3×** to **1.7×** on the sparse attention computation, compared to the TidalDecode approach under the same token budgets. This is mainly due to LessIsMore's unified token selection design, which is more kernel-friendly for GQA-based models. More specifically, methods like TidalDecode/Quest require more KV loading per KV group when different query heads can select a different set of tokens.

*Table 2.* End-to-end single-step decoding latency (in ms) with a 2K token budget on DeepSeek-R1-Distilled-LLaMA-8B using the SGLang + FlashInfer serving stack (lower is better).

| Method (2K) | 16K | 32K | 64K |
|---|---|---|---|
| LessIsMore (Ours) | **23.0** | **23.4** | **24.1** |
| TidalDecode | 24.3 | 24.7 | 25.4 |
| Quest | 24.2 | 24.4 | 24.8 |
| Baseline (Full Attention) | 25.3 | 28.4 | 34.4 |

In Table 2, we also report end-to-end decoding latency when integrating LessIsMore and other sparse attention baselines into SGLang (Zheng et al., 2024), a modern serving stack. All methods were evaluated using the DeepSeek-R1-Distilled-LLaMA-8B model on a single NVIDIA A5000 GPU. For each method, we measure the per-token decoding latency under a token budget of 2K and context lengths of 16K, 32K, and 64K tokens. The backend attention computation is executed through SGLang with FlashInfer kernels, providing a unified and optimized execution environment for all baselines. Across all context lengths, LessIsMore achieves consistently lower latency than both baselines.

**Additional Experiments.** The appendix includes ablations on recency-window ratio (Section A.1.2) and generation length inflation of different sparse attention methods (Section A.1.3), kernel-level efficiency and memory profiling (Section A.3.1), LongBench and long-context retrieval results for GQA- (Tables 9 and 11) and MHA-based models (Table 10), and analysis of CUSA layer placement and recency locality in reasoning (Sections A.6 and A.7).

## 5. Related Work

**Sparse attention and KV cache compression.** Sparse attention reduces the computational and memory cost of long-context inference by restricting attention to a subset of tokens (Yang et al., 2024; Tang et al., 2024). Existing approaches fall into two categories. Eviction-based methods permanently discard tokens from the KV cache according to heuristic criteria, maintaining a compact cache throughout generation (Xiao et al., 2023; Zhang et al., 2023; Li et al., 2024; Adnan et al., 2024). Selection-based methods retain the full cache but dynamically select tokens to attend at each step, often using head-specific top-$k$ scores (Yang et al., 2024; Tang et al., 2024; Hao et al., 2025; Liu et al., 2024). While effective for retrieval and summarization, both paradigms struggle on reasoning workloads, where local selection errors accumulate over long generations.

**Sparse attention for reasoning.** Recent reasoning models exploit test-time scaling, where increasing generation length is often more effective than enlarging model size (Wei et al., 2023; DeepSeek-AI, 2025). However, the long-horizon nature of reasoning poses challenges for sparse attention: prior methods either incur substantial accuracy degradation under low token budgets (Yang et al., 2024; Tang et al., 2024; Cai et al., 2025) or rely on expensive post-training adaptations to recover accuracy (Gao et al., 2025), often further increasing generation length.

## 6. Conclusion

We presented LessIsMore, a training-free sparse attention mechanism tailored for long-horizon reasoning. Our key insight is that token importance in reasoning is a global and stable property, enabling sparse attention to move beyond head-local, short-sighted selection strategies. By enforcing cross-head unified token selection and preserving recent context, LessIsMore mitigates error accumulation and avoids the reasoning length inflation observed in prior methods. Extensive evaluations show that LessIsMore preserves lossless reasoning accuracy at high sparsity—e.g., achieving full accuracy on AIME-24 with a 2K token budget—while significantly improving efficiency. With optimized kernels, LessIsMore delivers up to $1.6\times$ end-to-end decoding speedup over full attention and up to $1.72\times$ faster

sparse attention computation compared to state-of-the-art baselines. Overall, our results demonstrate that explicitly exploiting global attention structure is critical for efficient and accurate sparse attention in reasoning workloads, and suggest a promising direction for scaling reasoning models without sacrificing performance.

## Acknowledgements

We thank the Princeton Language Institute (PLI) for providing the compute resources that supported this work.

## Impact Statement

This paper presents work whose goal is to advance the efficiency of inference for large reasoning models. By reducing decoding-time compute and memory access without retraining, LessIsMore lowers the cost and energy footprint of deploying reasoning models, which we view as a positive societal effect. Our method is training-free, model-agnostic, and does not introduce new data, capabilities, or modalities beyond those already present in the underlying models; consequently, we do not foresee additional ethical concerns or risks of misuse beyond those inherent to the pretrained reasoning models on which LessIsMore is applied.

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

## A. Appendix

This appendix provides supporting analyses and extended results for LessIsMore. We first present ablation studies (Section A.1) analyzing the effectiveness of unified cross-head aggregation on GQA models (Section A.1.1) and the impact of the recent-window ratio on attention recall and correctness (Section A.1.2). We then report comprehensive efficiency evaluations, including end-to-end decoding performance in the SGLang serving stack (Section A.3.2) and kernel-level analyses of FLOPs, memory transfer, and latency (Section A.3.1). Additional results cover an empirical variance analysis of the Pass@1 evaluation protocol (Section A.2), reasoning accuracy across multiple model sizes (Tables 7 and 8), long-context performance on Needle-in-the-Haystack and LongBench benchmarks (Tables 9 and 11), the choice of re-selection layers (Section A.6), and an empirical study of recency locality in the reasoning process (Section A.7).

### A.1. Ablation Study

#### A.1.1. EFFECTIVENESS OF LESSISMORE'S AGGREGATION ON GQA

To assess whether LessIsMore's unified selection generalizes to GQA models, we compare three aggregation strategies on Qwen3-8B with shared KV heads: **LessIsMore**, **Randomized Top-k**, and **Head-to-Head** (Figure 7). When selection is applied at all decoding layers, locally optimized schemes such as Randomized Top-k appear competitive. However, when selection is reduced to only Layer 2 (a more realistic low-frequency setting), these local heuristics fail to generalize, leading to substantially lower attention recall. In contrast, LessIsMore maintains strong recall in both settings, demonstrating that a globally consistent cross-head selection strategy is significantly more robust than layer-specific or head-specific methods. This underscores the importance of unified aggregation for stable token importance estimation under sparse selection.

#### A.1.2. EFFECT OF RECENT WINDOW RATIO

We analyze how the recent-window ratio $r$ affects attention recall and correctness on AIME-24 under a 4K token budget (Figure 8). Only configurations that combine a recent window with Cross-Head Selection (25%, 50%, 75%) successfully solve the task. Using only recent tokens ($r = 100\%$) yields the lowest recall by discarding essential long-range context. TidalDecode improves recall yet still fails to reach the correct answer, and using Cross-Head Selection with 0% recent further improves recall but likewise fails. Introducing even a modest recent window consistently boosts recall. The 25% configuration—corresponding to the LessIsMore design—achieves the highest recall across the generation,

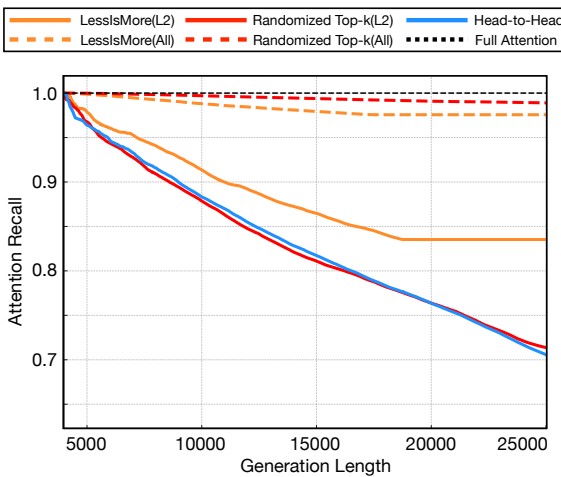

*Figure 7.* The Top-4K attention recall of different selection schemes applied only on Layer 2(L2) or all decoding layers(All). (1) **LessIsMore**: our unified top-k selection across all attention heads with 25% tokens for recency window, (2) **Randomized Top-k**: random application of one query head's top-k tokens to the entire KV group, and (3) **Head-to-Head**: direct utilization of top-k tokens for each individual attention head

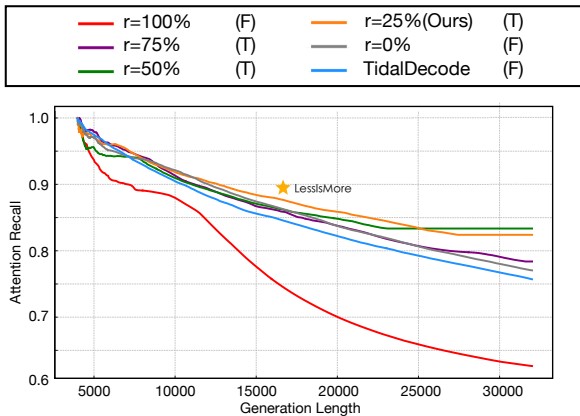

*Figure 8.* Ablation study on the impact of varying static recency window ratio $r$ in LessIsMore ($\star$) on the AIME-24 reasoning task, using a token budget of 4K and generation length up to 32K tokens on Qwen3-8B. LessIsMore corresponds to the 25% recent setting combined with Cross-Head Selection, (labeled with ($\star$)). We compare it against alternative recent window ratios, the 100% recent baseline (i.e., using only recent tokens), and TidalDecode.

validating the choice of allocating a small fraction of the token budget to recent tokens.

### A.1.3. GENERATION LENGTH ANALYSIS UNDER SPARSE ATTENTION

Sparse attention methods exhibit a concerning tendency that extends generation lengths on reasoning tasks, as demonstrated in Table 3 and corroborated by prior research (Gao et al., 2025). This phenomenon reflects the accumulation of selection errors discussed in Section 1, where imprecise

*Table 3.* AIME-24 accuracy (%) and average generation length (K tokens) on Qwen3-8B. Sparse methods often extend generation due to selection errors. LessIsMore maintains lengths close to Full Attention while achieving best accuracy.

| Method | K=2000 | | K=4000 | | K=6000 | |
|---|---|---|---|---|---|---|
| | Acc | Len | Acc | Len | Acc | Len |
| Quest | 18.2 | 30.0 | 46.7 | 22.9 | 49.6 | 19.6 |
| SeerAttention-r | 58.2 | 19.8 | 71.4 | 16.3 | 74.1 | 15.3 |
| TidalDecode | 53.3 | 17.4 | 70.0 | 16.9 | 71.3 | 15.9 |
| LessIsMore (Ours) | **73.8** | **15.8** | **75.8** | **14.8** | **76.7** | **15.1** |
| Full Attention | 74.5 | 14.8 | 74.5 | 14.8 | 74.5 | 14.8 |

token retention forces models into inefficient reasoning patterns that compromise both accuracy and computational efficiency.

Table 3 presents the average generation lengths of different approaches under various token budgets on AIME-24 with 16 sampled answers per problem using Qwen3-8B. Under restrictive token budgets (K=2000), existing methods generate substantially longer sequences compared to full attention: Quest, SeerAttention-r and TidalDecode each generate 30.0K, 19.8K, and 17.4K tokens, representing 103%, 34%, and 18% increases respectively over the full attention baseline of 14.8K tokens. These extended sequences indicate that sparse attention errors accumulate over time and may force models to engage in a redundant reasoning process. In contrast, LessIsMore maintains generation lengths closely aligned with full attention across all token budgets. At K=4000, LessIsMore generates the same number of tokens as full attention does while achieving better accuracy. Meanwhile, even with a token budget of 6K, TidalDecode obtains a significant lower accuracy and generates 15.9K tokens. Combining with the average decoding latency in Figure 6a, LessIsMore achieves a $1.16\times$ end-to-end speedup compared to TidalDecode.

Since inaccurate token selection leads to extended generation lengths, attention recall serves as an indicator of both selection accuracy and computational efficiency. Therefore, evaluating attention recall dynamics throughout generation becomes more crucial for assessing sparse attention methods on reasoning tasks.

### A.2. Reasoning Evaluation Variance Analysis

To demonstrate the variance in reasoning evaluation, we generate a pool of 512, 64, and 32 answers per problem in AIME-24/25, GPQA, and MATH500 with Full Attention on Qwen3-8B, respectively. We calculate the variance of different sample sizes in Table 4 by sampling outputs from the pool and computing the variance of final accuracy over 100 runs, where it shows the evaluated AIME-24/25, GPQA, and MATH500 exhibit minimal variance when evaluated on

*Table 4.* The variance of AIME-24 accuracy on Qwen3-8B with different sample sizes over 100 passes. The sampled variance of AIME-24/25, MATH500, and GPQA stay minimal ($<0.6$) with 64, 8, and 16 samples per problem.

| Model (Task) | Task / Sample Size | 8 | 16 | 64 |
|---|---|---|---|---|
| Qwen-3-8B (Variance) | AIME-24 | ±1.54 | ±0.98 | ±0.56 |
| | AIME-25 | ±1.76 | ±1.12 | ±0.58 |
| | MATH500 | ±0.14 | ±0.11 | ±0.05 |
| | GPQA | ±0.59 | ±0.43 | ±0.20 |

sample size of 64, 16, and 8, respectively. These sample sizes are consequently the ones used in our main evaluation protocol (Section 4.1), confirming that the accuracy gaps reported in Figure 5 exceed evaluation noise.

*Table 5.* End-to-end TBT speed-up of LessIsMore on SGLang serving stack under different context lengths.

| Method | 16K | 32K | 64K |
|---|---|---|---|
| SGLang + LessIsMore-2K | 1.11 | 1.25 | 1.51 |
| SGLang + LessIsMore-4K | 1.09 | 1.22 | 1.48 |

*Table 6.* Kernel-level FLOP count, global-to-shared (G2S) memory transfer, memory consumption, and latency (2K budget, 16K context, DeepSeek-R1-Distilled-LLaMA-8B).

| Method | FLOPs | G2S | Mem | Lat. |
|---|---|---|---|---|
| LessIsMore (Ours) | 1.05M | 1.04MB | 8.38MB | 20.1$\mu$s |
| TidalDecode | 1.05M | 2.34MB | 8.38MB | 32.1$\mu$s |
| Quest/SeerAttn-R | 1.05M | 2.34MB | 8.38MB | 32.1$\mu$s |
| StreamingLLM | 1.05M | 1.04MB | 1.04MB | 20.1$\mu$s |
| Full Attention | 8.40M | 8.38MB | 8.38MB | 76.4$\mu$s |

## A.3. Efficiency Evaluation

### A.3.1. KERNEL-LEVEL FLOP, MEMORY TRANSFER, AND LATENCY ANALYSIS

To complement the end-to-end evaluations, we report kernel-level metrics that highlight the efficiency benefits of LessIsMore relative to other sparse attention methods. Using the DeepSeek-R1-Distilled-LLaMA-8B model with a token budget of 2K and a context length of 16K, we profile the FLOPs, global-to-shared memory data transfer, on-device memory consumption, and per-kernel latency for the attention computation. All measurements are obtained using FlashInfer as the backend attention kernel library. Table 6 shows that even if the FLOP count is identical across sparse attention methods, LessIsMore performs significantly less global-to-shared memory transfer than TidalDecode or Quest/SeerAttention-R due to its unified cross-head token selection, which reduces redundant KV loading across atten-

tion heads. This reduction directly contributes to our lower kernel latency. StreamingLLM achieves optimal FLOP and memory-transfer numbers but performs poorly on reasoning accuracy due to its static attention structure, making it less suitable for long-form reasoning tasks. The full-attention baseline incurs substantially higher FLOPs and memory movement, resulting in much higher latency.

### A.3.2. END-TO-END SPEEDUP ON INFERENCE ENGINE

To evaluate the practical impact of LessIsMore under modern serving infrastructures, we additionally measure end-to-end decoding performance when integrating our method into the SGLang serving stack (Zheng et al., 2024), which is built on top of the FlashInfer attention kernel library. Since FlashInfer is also used by vLLM (Kwon et al., 2023) and provides state-of-the-art fused attention kernels, this experiment reflects realistic deployment conditions. All baseline methods, including TidalDecode, are also implemented using FlashInfer kernels to ensure a fair comparison.

Table 5 reports the time-between-token (TBT) speed-up achieved when applying LessIsMore with different token budgets under 16K, 32K, and 64K context lengths. Across all settings, LessIsMore consistently accelerates end-to-end decoding, reaching up to **1.51×** speed-up at 64K context length.

## A.4. Reasoning Evaluation

### A.4.1. REASONING EVALUATION RESULTS ON QWEN3-4/8/14B, AND DEEPSEEK-R1-DISTILL-LLAMA-8B

In Table 7 and Table 8, we record Pass@1 accuracy plotted in Figure 5 with more sparse attention baselines and token budgets.

## A.5. Long-Context Evaluation

### A.5.1. NEEDLE-IN-THE-HAYSTACK

Table 9 shows that on both non-reasoning models Llama-3-8B-Instruct-Gradient-1048k (AI, 2024a) and Llama-3.1-8B-Instruct (AI, 2024b), LessIsMore maintains strong long-context retrieval performance, consistently outperforming Quest and matching or exceeding TidalDecode even at very small token budgets. LessIsMore and TidalDecode both apply Layer 13 as re-selection layer. Remarkably, LessIsMore achieves full Needle-in-the-Haystack accuracy with only 32–128 tokens with up to 100K context (0.1–0.3% of the input), demonstrating that LessIsMore remains effective for long-context retrieval tasks and generalizes well beyond reasoning-oriented models.

### A.5.2. EVALUATION OF LONGBENCH

Evaluation of LessIsMore, Quest, and TidalDecode on Long-Bench datasets (Bai et al., 2023) is shown in Table 11. As a result, LessIsMore consistently achieves higher average F1 across five LongBench datasets MultiFieldQA, Qasper, HotpotQA, TriviaQA, PassageRetrival, matching or surpassing the Full Attention baseline while using only a 4K token budget and even achieve the highest average score. These results highlight that LessIsMore not only preserves accuracy in complicated reasoning tasks but also exhibits potential on solving long-context tasks.

### A.5.3. GENERALIZATION OF LESSISMORE ON MHA

LessIsMore is not limited to GQA-based architectures; its unified cross-head token selection strategy naturally extends to standard MHA-based models as well. To demonstrate this generality, we apply LessIsMore to LongChat-7B-v1.5-32k (Li et al., 2023)—an MHA-based long-context model—and evaluate performance on the 10k-context Needle-in-the-Haystack benchmark. As shown in Table 10, LessIsMore consistently matches or surpasses strong selection-based baselines (Quest, TidalDecode) and significantly outperforms eviction-based approaches, achieving full accuracy with only a 256-token budget. These results highlight that LessIsMore captures global token-importance patterns that remain effective even without GQA structure, underscoring its robustness and architectural generality.

## A.6. Choice of Optimal Re-Selection in LessIsMore

Following the procedure of choosing optimal re-selection layer of TidalDecode (Yang et al., 2024), we conduct a simple 5K-context-length needle-in-the-haystack test with PG-19-mini (Rae et al., 2019) on TidalDecode with each evaluated model. With a token budget of 256, Layer 12 on Qwen3-8B/14B and DeepSeek-R1-Distill-Llama-8B provides the highest accuracy while Layer 12 and Layer 20 on Qwen-4B offer very similar performance. Moreover, prior work has found that in the same model family, the optimal re-selection layer is similar. For Qwen3, we validate that Layer 12 is an important layer. To demonstrate the generalization of our approach on different models, we choose different re-selection layers for different Qwen3 models. In this paper's experiments Section 4, we apply the same re-selection layer on TidalDecode and LessIsMore for a fair comparison - Layer 12 and Layer 20 for Qwen3-8B/14B/DeepSeek-R1-Distill-Llama-8B and Qwen3-4B, respectively. We provide a table Table 12 summarizing the re-selection layer used for each model in this paper.

*Table 13.* Impact of re-selection layer on AIME-24 accuracy (Qwen3-8B, 64 samples). Layer 12 consistently achieves best performance, validating the needle-in-haystack search procedure.

| Method | K=2K | K=4K | K=6K | K=8K |
|---|---|---|---|---|
| LessIsMore+(None) | 58.02 | 63.33 | 66.17 | 70.00 |
| LessIsMore+L5 | 53.33 | 62.60 | 70.31 | 74.68 |
| LessIsMore+L18 | 71.67 | 72.91 | 73.33 | 74.79 |
| LessIsMore+L30 | 63.23 | 65.83 | 70.10 | 75.27 |
| **LessIsMore+L12 (Ours)** | **73.00** | **75.56** | **76.45** | **76.67** |
| Full Attention | 74.48 | 74.48 | 74.48 | 74.48 |

As shown in Table 13, the choice of re-selection layer has a clear and measurable impact on accuracy across all token budgets. Earlier or later layers (e.g., L5, L18, L30) consistently underperform compared to L12, indicating that re-selection must occur at a layer that balances sufficient semantic abstraction with stable attention patterns. LessIsMore+L12 achieves the highest accuracy in every budget setting, matching or exceeding the Full Attention baseline. These results confirm two key points: (1) the re-selection layer is indeed critical for sparse attention performance, and (2) the needle-in-the-haystack search used to identify the optimal TidalDecode layer (L12 for Qwen3-8B) reliably predicts the best re-selection position for LessIsMore as well. This validates our use of the same re-selection layer for TidalDecode and LessIsMore to ensure a fair and methodologically sound comparison.

## A.7. Recency Locality in Reasoning Process

Figure 9 and Figure 10 depicts the ground-truth distribution of selected tokens in GPQA and AIME-25 datasets with the

*Table 7.* LessIsMore vs. Full Attention accuracy (%) on AIME-24/25 with 64 sampled answers per problem. LessIsMore matches or exceeds Full Attention across nearly all configurations. Results exceeding Full Attention in **bold**.

| Model | Method | AIME-24 | | | | AIME-25 | | | |
|---|---|---|---|---|---|---|---|---|---|
| | | **2K** | **4K** | **6K** | **8K** | **2K** | **4K** | **6K** | **8K** |
| Qwen3-4B | LessIsMore | **71.48** | **73.03** | **74.37** | **73.12** | 62.87 | 65.94 | **66.56** | **66.46** |
| | *Full Attn* | | | 71.25 | | | | 66.41 | |
| Qwen3-8B | LessIsMore | 73.00 | **75.56** | **76.45** | **76.67** | 65.24 | **70.31** | **68.00** | **68.02** |
| | *Full Attn* | | | 74.48 | | | | 67.86 | |
| Qwen3-14B | LessIsMore | 78.58 | **80.39** | **80.19** | **80.10** | 66.56 | **70.59** | 70.48 | **70.52** |
| | *Full Attn* | | | 79.79 | | | | 70.52 | |
| DeepSeek-8B | LessIsMore | 43.22 | **44.28** | **45.84** | **45.10** | 29.93 | **31.91** | **31.93** | **31.67** |
| | *Full Attn* | | | 44.16 | | | | 30.83 | |

*Table 8.* LessIsMore vs. Full Attention accuracy (%) on MATH500 and GPQA-Diamond with 8 and 16 sampled answers per problem, respectively. LessIsMore matches or exceeds Full Attention across nearly all token budgets. Results exceeding Full Attention in **bold**.

| Model | Method | MATH500 | | | | GPQA-Diamond | | | |
|---|---|---|---|---|---|---|---|---|---|
| | | **1K** | **2K** | **4K** | **6K** | **1K** | **2K** | **4K** | **6K** |
| Qwen3-4B | LessIsMore | 92.50 | **94.12** | **94.16** | **94.22** | **56.48** | **56.23** | **56.84** | **56.64** |
| | *Full Attn* | | | 93.93 | | | | 56.19 | |
| Qwen3-8B | LessIsMore | 93.35 | **94.55** | **94.45** | 94.42 | 58.62 | **60.65** | **61.58** | **61.11** |
| | *Full Attn* | | | 94.43 | | | | 60.54 | |
| Qwen3-14B | LessIsMore | 94.40 | 94.85 | **95.14** | **95.12** | 63.42 | 63.61 | **64.85** | **65.15** |
| | *Full Attn* | | | 95.10 | | | | 64.02 | |
| DeepSeek-8B | LessIsMore | **86.90** | **88.15** | **88.75** | **88.54** | 35.74 | 39.11 | **43.08** | **43.31** |
| | *Full Attn* | | | 85.45 | | | | 42.80 | |

recency locality across attention heads being highlighted. The recency locality is a pattern persistent throughout the thinking process of LRMs regardless of token budgets and tasks. Notably, the size of the highlighted range stays relatively consistent as the model generates more tokens but grows proportionally to the token budgets. This reinforces the effectiveness of design choice of LessIsMore. Stable Recency Window in Section A.1.2 leverages the nature of reasoning process and captures the most critical tokens by allocating a fixed ratio of token budgets for most recent tokens.

*Table 9.* Results of 10K-, 32K-, and 100K-context Needle-in-the-Haystack tests on non-reasoning models Llama-3-8B-Instruct-Gradient-1048k (AI, 2024a) and Llama-3.1-8B-Instruct (AI, 2024b) using the PG-19-mini dataset (Rae et al., 2019). Across both models, LessIsMore consistently matches or surpasses TidalDecode and Quest, demonstrating that its unified token selection effectively captures key information even in purely long-context retrieval settings. Notably, LessIsMore achieves full accuracy with only 32, 32, and 128 tokens for 10K-, 32K-, and 100K-context tasks, corresponding to just 0.3%, 0.1%, and 0.1% of the total input lengths, respectively.

| Model (context length) | Method / Budget | K=32 | K=64 | K=128 | K=256 | K=512 |
|---|---|---|---|---|---|---|
| Llama-3-8B (10K) | Quest | 74% | 84% | 99% | 98% | **100%** |
| | TidalDecode | 88% | 98% | **100%** | **100%** | **100%** |
| | LessIsMore(Ours) | **100%** | **100%** | **100%** | **100%** | **100%** |
| Llama-3-8B (100K) | Quest | 38% | 50% | 65% | 87% | 98% |
| | TidalDecode | 86% | 92% | **100%** | **100%** | **100%** |
| | LessIsMore(Ours) | **98%** | **100%** | **100%** | **100%** | **100%** |
| Llama-3.1-8B (10K) | Quest | 74% | 86% | 94% | **100%** | 98% |
| | TidalDecode | **100%** | **100%** | **100%** | **100%** | **100%** |
| | LessIsMore(Ours) | **100%** | **100%** | **100%** | **100%** | **100%** |
| Llama-3.1-8B (32K) | Quest | 78% | 88% | 92% | **100%** | **100%** |
| | TidalDecode | **98%** | **100%** | **100%** | **100%** | **100%** |
| | LessIsMore(Ours) | **100%** | **100%** | **100%** | **100%** | **100%** |

*Table 10.* Results of the 10k-context Needle-in-the-Haystack test on the MHA-based LongChat-7B-v1.5-32k model (Li et al., 2023). This experiment demonstrates that LessIsMore generalizes beyond GQA-based architectures, achieving equal or superior accuracy compared to selection-based baselines such as Quest (Tang et al., 2024) and TidalDecode (Yang et al., 2024), and substantially outperforming eviction-based approaches including H2O (Zhang et al., 2024), TOVA (Oren et al., 2024), and StreamingLLM (Xiao et al., 2023). Notably, LessIsMore reaches full accuracy with only a 256-token budget, matching full-attention performance under extreme sparsity.

| Method / Budget | K=32 | K=64 | K=128 | K=256 | K=512 |
|---|---|---|---|---|---|
| H2O | 0% | 1% | 1% | 1% | 3% |
| TOVA | 0% | 1% | 1% | 3% | 8% |
| StreamingLLM | 1% | 1% | 1% | 3% | 5% |
| Quest | 65% | **99%** | **99%** | 99% | **100%** |
| TidalDecode | 73% | 92% | 98% | 99% | **100%** |
| LessIsMore(Ours) | **92%** | 98% | 98% | **100%** | **100%** |

*Table 11.* Performance comparison on five LongBench datasets MultiFieldQA(MFQA), Qasper(Qasp), HotpotQA(HotQA), TriviaQA(TrQA), PassageRetrieval-en(PRe), testing capabilities of long-context retrieval, multi-hop Q&A, multi-document comprehension, and structured information integration of each approach. The highest F1-score or accuracy for each task is in bold.

| Method (K) | MFQA | Qasp | HotQA | TrQA | PRe | Avg |
|---|---|---|---|---|---|---|
| Full Attention | 30.76 | **14.56** | 11.50 | 86.56 | 77.00 | 44.08 |
| Quest (1024) | 26.21 | 12.19 | 10.75 | 83.47 | 63.84 | 39.29 |
| TidalDecode (1024) | 28.57 | 11.11 | 9.82 | 79.78 | 75.17 | 40.89 |
| LessIsMore(Ours) (1024) | 29.87 | 14.20 | 12.04 | **87.42** | 75.58 | 43.82 |
| Quest (4096) | 28.92 | 13.63 | 12.15 | 85.91 | 72.50 | 42.62 |
| TidalDecode (4096) | **30.94** | 13.85 | **13.71** | 86.30 | 78.00 | 44.56 |
| LessIsMore(Ours) (4096) | 30.90 | 14.34 | 12.58 | 87.06 | **79.00** | **44.78** |

*Table 12.* Re-selection layers used during evaluation, with model family and task type annotations.

| Model | Family | Attn Type | Reasoning | Re-selection Layer |
|---|---|---|---|---|
| LongChat-7B-v1.5-32k | Llama2 | MHA | ✗ | 7 |
| Llama-3-8B | Llama3 | GQA | ✗ | 13 |
| Llama-3.1-8B | Llama3 | GQA | ✗ | 13 |
| DeepSeek-R1-Distill-Llama-8B | Llama3 | GQA | ✓ | 12 |
| Qwen3-4B | Qwen3 | GQA | ✓ | 20 |
| Qwen3-8B | Qwen3 | GQA | ✓ | 12 |
| Qwen3-14B | Qwen3 | GQA | ✓ | 12 |

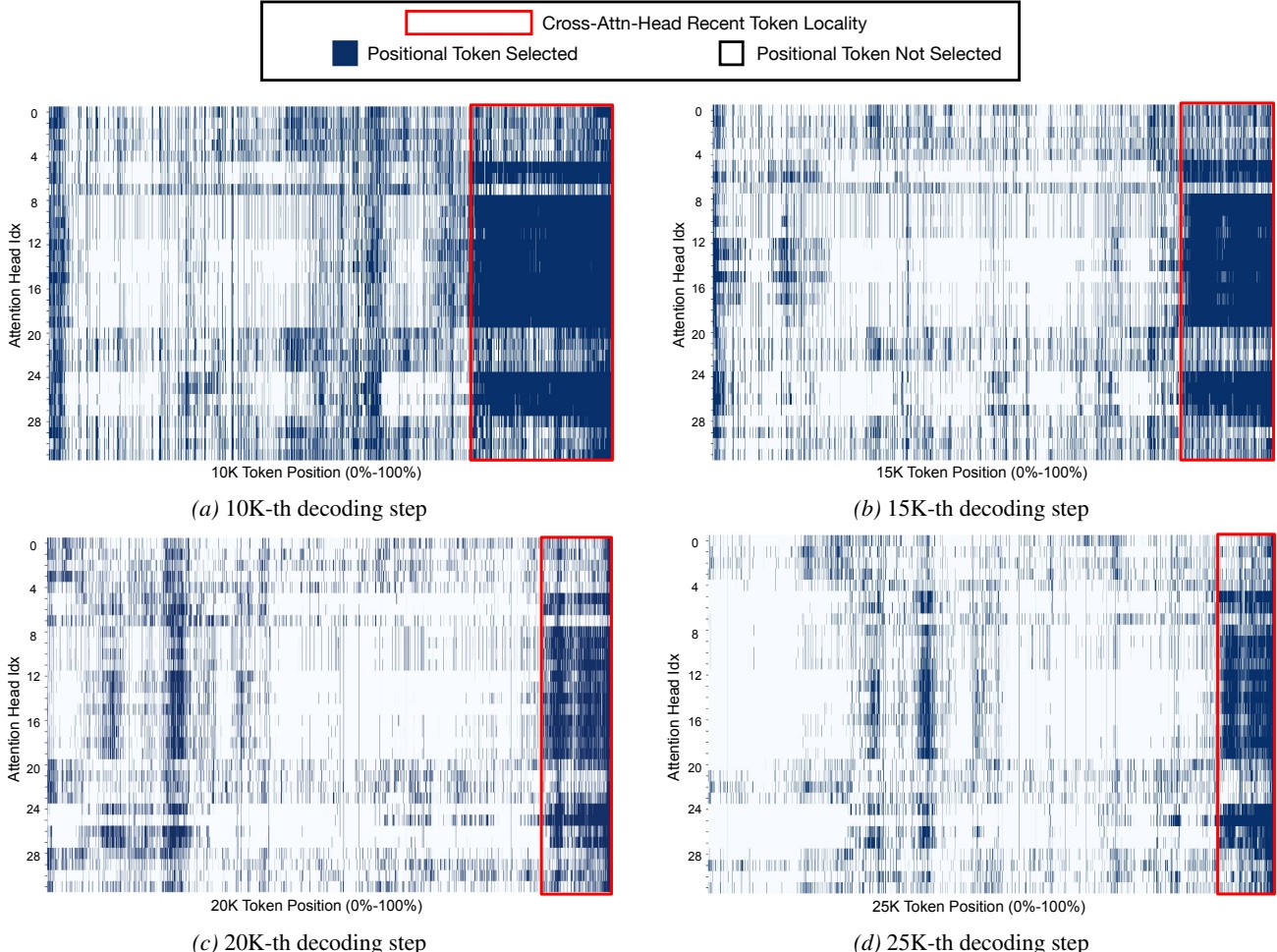

*(a)* 10K-th decoding step

*(b)* 15K-th decoding step

*(c)* 20K-th decoding step

*(d)* 25K-th decoding step

*Figure 9.* The distribution of the ground-truth top-4K tokens across all attention heads at 10K-, 15K-, 20K-, and 25K-th decoding step at Layer 4 on AIME-24 with Qwen3-8B. We enclose the highly overlapped area of attention heads within the same KV group with red, which forms a most recent window across all decoding steps

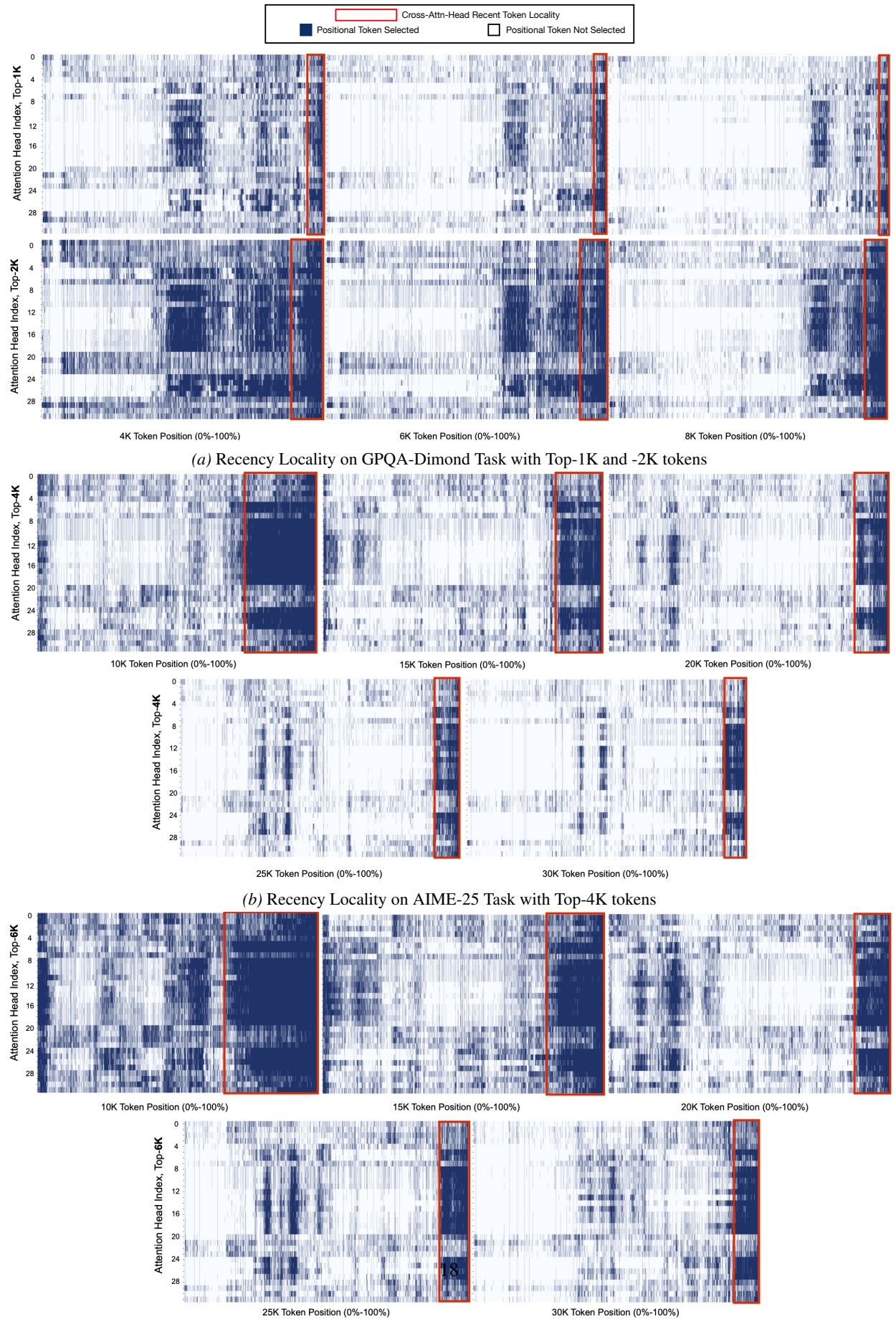

*(a)* Recency Locality on GPQA-Dimond Task with Top-1K and -2K tokens

*(b)* Recency Locality on AIME-25 Task with Top-4K tokens

*(c)* Recency Locality on AIME-25 Task with Top-6K tokens

*Figure 10.* The ground-truth distribution of selected tokens across different decoding steps and various reasoning tasks on the Layer 4 of