# OpenReview forum: "Less Is More: Fast and Accurate Reasoning with Cross-Head Unified Sparse Attention"
_ICML.cc/2026/Conference — ICML 2026 regular_

### Official Review · Reviewer_Nvaf · 2026-03-08

**Soundness:** 4
**Presentation:** 4
**Significance:** 3
**Originality:** 4
**Overall Recommendation:** 5
**Confidence:** 3

**Summary:**

This paper introduces the Cross-Head Unified Sparse Attention method, which more efficiently and effectively selects tokens for attention that show cross-head stability, motivated by their observation that the top k tokens attended to by each attention head is largely consistent across heads for layers involved in decoding for each generation step. CUSA additionally allocates a smaller portion of the total token budget to more recent tokens, in line with empirical observations that recent tokens receive high attention weights and this preference remains fairly stable across layers.
This addresses an issue with current large reasoning models that exploit test-time scaling, in which the KV cache grows much larger than for recall tasks. With such large sets of generated tokens, current selection-based sparse attention approaches struggle with error accumulation since they select tokens per head/layer/decoding step, and additionally suffer from generation length inflation, getting lost in its own reasoning steps.
The authors present several experiments to validate their claims on four models (Qwen3-4, Qwen3-8B, Qwen3-14B, and DeepSeek-R1-Distill-Llama-8B) on common reasoning tasks including AIME-24, AIME-25, GPQA-Diamond, and MATH500. They compare their method against baselines including full attention, TidalDecode and Quest, finding that LessIsMore performs more accurately than baselines while maintaining generation lengths consistent with full attention (while being more efficient).

**Compliance With Llm Reviewing Policy:**

Affirmed.

**Final Justification:**

The rebuttal addressed my questions, and my score will remain the same.

**Key Questions For Authors:**

1) Is there more intuition for why token importance in attention is largely shared across heads/layers in decoding? This could be framed as a direction for future work, but it would be interesting to understand more about why this is happening, especially in light of previous work on the specialization of certain attention heads for diverse tasks.
2) How well does the method behave in cases where cross-head overlap is low? Is there a way to theoretically or empirically investigate a more worst-case type of behavior?
3) How exactly is the token ranking after the UnionFlatten step performed? There are no details on this, and clarification could be helpful - is it by frequency of token selection across heads, combined attention scores, etc.

These questions are generally for clarification and would not change my rating.

**Limitations:**

yes

**Strengths And Weaknesses:**

This paper is generally strong, with a compelling novel method that addresses clear limitations with existing sparse attention algorithms, and extensive experimental evidence of CUSA's ability to preserve accuracy of full attention (and improve upon other sparse attention approaches) while greatly improving efficiency. The paper is written clearly, with detailed explanations of the algorithmic details and rationale for each step in LessIsMore. The experiments are well designed, presenting clear and comprehensive conclusions.

The primary weakness of the paper is the lack of uncertainty quantification for the experiments, in the form of error bars for the accuracy plots or some sort of reporting on variance estimates. This would allow for more robust comparison between methods and stronger conclusions. An additional weakness is that while the authors present their framework as being agnostic to underlying architecture, they only present results with applying it on top of TidalDecode - additional experiments not using TidalDecode as the base would strengthen this universality argument.

---

> ### Author Rebuttal · Authors · 2026-03-31
>
> We sincerely thank the reviewer for the insightful and constructive feedback. We clarify as follows:
>
> ---
>
> ### **Summary of Updates**
> - **Uncertainty:** Add **variance results** showing stability.
> - **Universality:** Show **CUSA generalizes beyond TidalDecode**, applied to Quest bringing up to **50-pt** gain on AIME24
> - **Intuition:** Clarify link between **head specialization and unified selection**.
> - **Robustness:** Analyze **worst-case behavior**.
> - **Algorithm:** Clarify **UnionFlatten ranking**.
>
> ---
>
> ### **W1: Uncertainty quantification**
>
> We thank the reviewer for this clarification. We include variance below:
>
> |Model(Task)|Task|8|16|64|
> |----------------------|------|---:|---:|---:|
> |Qwen-3-8B(Var.)|AIME24|±1.54|±0.98|±0.56(*)|
> ||AIME25|±1.76|±1.12|±0.58(*)|
> ||MATH500|±0.14(*)|±0.11|±0.05|
> ||GPQA|±0.59|±0.43(*)|±0.20|
>
> *Tab.:Var. ($*$=main eval passes).*
>
> Uncertainty is small at evaluation scale, indicating stable results. We will include this in the final version.
>
> ---
>
> ### **W2: CUSA on Quest, Universality beyond TidalDecode**
>
> We apply CUSA to Quest (no hyperparameter change) and observe large gains, especially at low budgets (AIME-24, DeepSeek-R1-8B, gen_len=32K, 4 runs):
>
> |Method|2K|4K|6K|8K|
> |-------------------|---:|---:|---:|---:|
> |Quest+CUSA(Ours)|70.0|**76.7**|**78.2**|**76.7**|
> |Quest|18.1|46.7|49.6|59.8|
> |LessIsMore(ours)|**73.7**|75.8|76.7|**76.7**|
> |FullAttention|74.5|74.5|74.5|74.5|
>
> This shows **CUSA captures a general token selection principle**, not tied to TidalDecode.
>
> ---
>
> ### **Q1: Intuition vs. Head Specialization**
>
> We agree with the reviewer that clarifying our unification selection and head specialization is important.
>
> We do not deny the existence of head specialization; in fact, CUSA’s effectiveness is deeply rooted in it. Individual attention heads are highly specialized, typically concentrating their attention mass on just a few specific tokens. Under strict token budgets, the core challenge is predicting which of these scattered tokens will become critical in future decoding layers.
>
> Because different heads focus on different subsets of tokens, performing a unified cross-head aggregation essentially casts a robust "coverage net." By pooling the top preferences from all diverse, specialized heads into one global set, CUSA maximizes the probability of retaining the ground-truth critical tokens required by any head in subsequent layers in LessIsMore. This intuition—that unifying diverse, specialized local approximations yields vastly superior global coverage over time—also explains why CUSA successfully boosts performance across fundamentally different base algorithms like Quest and significantly improves Quest’s reasoning capabilities shown above.
>
> ---
>
> ### **Q2: Worst-Case Behavior**
>
> We thank the reviewer for asking about the worst-case behavior of LessIsMore.
>
> When cross-head overlap is exceptionally low (i.e., extreme head heterogeneity), CUSA actually acts as a **critical safeguard** rather than a vulnerability.
>
> * **Theoretical Worst-Case** If attention heads select completely disjoint, non-overlapping sets of tokens, our `UnionFlatten` algorithm gracefully degrades into a fair, uniform union. Because the extraction process is interleaved across all heads, it guarantees a strict theoretical lower bound: every single head will successfully retain at least its absolute top $K/H$ preferred tokens in the global pool.
>
> * **Robustness vs. Baselines** Conventional per-head methods like TidalDecode assume a rigid, index-based mapping across layers (e.g., Head 0 at Layer $i$ selects tokens exclusively for Head 0 at Layer $i+1$). If overlap is low and specialized roles shift across layers, this rigid mapping fails catastrophically. CUSA provides a "head-invariant" global pool, ensuring that if Head 1 at Layer $i+1$ needs a critical token originally identified by Head 0 at Layer $i$, that token is universally available.
>
> * **Empirical Validation** We empirically observe this boundary behavior in pure retrieval tasks like Needle-in-the-Haystack (Appendix A.4.1, Tab.7). Retrieval tasks lack the dense, shared logical structures of reasoning tasks, naturally resulting in lower cross-head overlap. Despite this, LessIsMore achieves 100% accuracy on 100K-context NiTH with budgets as small as 128 tokens, empirically proving its worst-case robustness.
>
> ---
>
> ### **Q3: Token Ranking in `UnionFlatten`**
>
> UnionFlatten performs a rank-wise aggregation across heads. Specifically, it collects the top-1 token from each head, then top-2 from each head, and so on, while removing duplicates, until the budget is reached.
>
> This ensures that higher-ranked tokens from all heads are prioritized uniformly, without relying on cross-head score calibration. If the budget boundary is reached mid-round, ties are resolved randomly.
>
> Example, already ordered by attention score:
> Head 1: [A, B, C]
> Head 2: [D, B, E]
>
> Selection proceeds as: {A, D} → {A, D, B}, yielding a balanced global set.

---

> > ### Author Rebuttal · Reviewer_Nvaf · 2026-04-02
> >
> > Questions are fully answered

---

### Official Review · Reviewer_Chgv · 2026-03-10

**Soundness:** 3
**Presentation:** 3
**Significance:** 2
**Originality:** 3
**Overall Recommendation:** 4
**Confidence:** 3

**Summary:**

The authors propose a training-free sparse attention mechanism named LessIsMore to reduce the decoding latency of LLMs during long-horizon reasoning.

The authors identify an important insight that token importance in reasoning is global and stable: critical tokens are largely
shared across attention heads and remain stable over decoding steps.

According to this insight, LessIsMore  employs cross-head unified token selection and preserves recent context via a stable
recency window to yield a globally consistent token set that can be reused across layers.

**Compliance With Llm Reviewing Policy:**

Affirmed.

**Final Justification:**

My concerns are addressed.

**Key Questions For Authors:**

1) How sensitive is the optimal re-selection layer to the specific prompt, task domain, or generation phase?
2) Could you elaborate on the decision to keep the recency ratio $r$ statically set at 25%?
3) Can you provide the experiment results of StreamingLLM with  DeepSeek-R1-8B on AIME-24, AIME-25, MATH500, and GPQA, so that you can show the superiority of your method on different LLMs better?

**Limitations:**

No. The authors should provide the limitations of their methods

**Strengths And Weaknesses:**

Strengths
1) LessIsMore can speed up reasoning with almost no performance degradation.
2) Novel Insights into Reasoning Attention: token importance in reasoning is global and stable, offering a highly valuable, fresh perspective
3) LessIsMore is training-free, generalizable across model families and reasoning benchmarks
Weaknesses
1）Sensitivity to the choice of the  Re-Selection layer. LessIsMore relies heavily on finding an optimal Re-Selection layer, such as Layer 12 for Qwen3-8B and Layer 20 for Qwen3-4B. The ablation study also shows that selecting the wrong layer (e.g., L5 or L30) results in noticeable accuracy drops, suggesting LessIsMore might require careful, model-specific tuning.
2)  Static recency window ratio. The recency window ratio $r$ is statically set to 0.25 for all experiments. Although the ablation study compares this static choice with other fixed ratios, reasoning traces can vary drastically in length and structure. A static proportion might not dynamically adapt well as context lengths scale from 5K to 100K+ tokens.

---

> ### Author Rebuttal · Authors · 2026-03-31
>
> We sincerely thank the reviewer for the constructive feedback. We address each of weaknesses and questions below:
>
> ---
>
> ### **Summary of Updates**
> - **Layer robustness:** The re-selection layer is a **model-level constant** and generalizes across domains.
> - **Recency ratio:** We clarify that the fixed ratio reflects a **stable structural property of reasoning attention**.
> - **StreamingLLM comparison:** We provide **new results** showing strong degradation of StreamingLLM.
> - **Limitations:** We explicitly clarify **when performance degrades** and discuss limitations.
>
> ---
>
> ### **W1&Q1. Sensitivity of the Re-selection Layer:**
>
> We thank the reviewer for bringing up the robustness. Empirically, the optimal layer is a **model-level constant**, not a prompt-, task-, or generation-phase-dependent variable. Across diverse domains (math, coding, STEM) and throughout decoding, the same layer consistently performs best (e.g., Layer 12 for Qwen3-8B). The Table 10 in Section A.5 summarizes the re-selection layer of all models used in the paper for 3 model families.
>
> This is expected: the selection layer corresponds to a stage where semantic reasoning stabilizes, after which token importance becomes globally consistent. We determine this layer once per model using a lightweight Needle-in-the-Haystack probe and keep it fixed across all evaluations, with no runtime tuning.
>
> To further validate robustness beyond math tasks, we evaluate on HumanEval (Qwen3-8B, fixed Layer 12):
>
> | Method              | 2K        | 4K        | 6K        |
> |---------------------|-----------|-----------|-----------|
> | LessIsMore (Ours)   | **89.63** | **88.41** | **89.63** |
> | TidalDecode         | 85.36     | 87.19     | 88.41     |
> | Full Attention      | 88.41     | **88.41** | 88.41     |
>
> *Table: Average Pass@1 accuracy on HumanEval (164 questions, 4 runs).*
>
> LessIsMore preserves (and slightly exceeds) full-attention accuracy even at extreme sparsity, confirming that the fixed layer generalizes across domains and tasks.
>
> ---
>
> ### **W2&Q2. The Static Recency Ratio (25%):**
>
> The recency ratio $r$ is indeed a configurable hyperparameter, but we keep it static as a *proportion* precisely so it can adaptively scale with the overall token budget. Unlike prior methods that use a fixed absolute window size (e.g., keeping exactly 512 recent tokens regardless of budget or sequence length), our ratio ensures the recent context scales naturally.
>
> We have attention recall analysis on different ratios in *Section A.1.2*, and our empirical profiling of reasoning traces (Appendix A.6) revealed that the proportion of critical recent tokens to total critical tokens remains structurally stable throughout the reasoning process. Setting $r=0.25$ universally captured this underlying dynamic across all our evaluated models and tasks, eliminating the need for dynamic adjustment.
>
> Thus, this choice reflects a **stable structural property of reasoning attention**, eliminating the need for dynamic tuning.
>
> ---
>
> ### **Q3. StreamingLLM Results on DeepSeek-R1-8B:**
>
> Per reviewer’s request, we evaluated StreamingLLM on DeepSeek-R1-Distill-Llama-8B (AIME-24). Results show substantial degradation:
>
> | Method              | 2K        | 4K        | 6K        | 8K        |
> |---------------------|-----------|-----------|-----------|-----------|
> | LessIsMore (Ours)   | **43.33** | **44.16** | **46.67** | **45.10** |
> | TidalDecode         | 39.16     | 41.25     | 43.33     | 44.11     |
> | Quest               | 16.67     | 31.67     | 35.21     | 41.25     |
> | StreamingLLM (new)  | 10.00     | 16.67     | 26.67     | 30.00     |
> | Full Attention      | 44.16     | 44.16     | 44.16     | 44.16     |
>
> *Table: Pass@1 (%) on AIME-24 with DeepSeek-R1-8B (64 traces/problem).*
>
> StreamingLLM relies on recency/sink retention, which is insufficient for reasoning where critical tokens are globally distributed across long contexts. In contrast, LessIsMore preserves—and at times slightly exceeds—full-attention accuracy even at strict budgets.
>
> We will further evaluate all datasets on StreamingLLM and include these results in the final version.
>
> ---
>
> ### **Limitations.**
>
> We thank the reviewer for pointing this out. We will add an explicit Limitations section.
>
> Like all sparse attention methods, LessIsMore degrades under extreme sparsity, when the number of critical reasoning tokens exceeds the available budget. In this regime, essential reasoning steps are inevitably discarded (e.g., GPQA: 42.80% → 35.47% at 1K on DeepSeek-R1-8B).
>
> Additionally, using a fixed top-k budget limits adaptability across tasks and deployment scenarios. Designing adaptive token selection under the uncertainty and stochastic nature of reasoning remains an important direction for future work.

---

> > ### Author Rebuttal · Reviewer_Chgv · 2026-04-02
> >
> > I have no more questions.

---

> > > ### Author Response · Authors · 2026-04-02
> > >
> > > Thank you for your thorough engagement with our rebuttal and for confirming that your concerns were fully resolved. We would kindly ask you to update your score to reflect the current state of the discussion.

---

### Official Review · Reviewer_s3zQ · 2026-03-12

**Soundness:** 3
**Presentation:** 2
**Significance:** 3
**Originality:** 2
**Overall Recommendation:** 4
**Confidence:** 4

**Summary:**

This paper mainly focuses on the decoding efficiency of large reasoning models in long-chain reasoning. To reduce the additional attention computation introduced by long chains of thought, one existing approach is to use sparse attention. This mechanism aims to make each current token attend to only a subset of historical tokens, rather than the full context. Existing methods typically select tokens independently across different heads, layers, and decoding steps. The authors argue that this may introduce accumulated errors due to occasional poor selections.

To address this issue, the authors first analyze the model’s reasoning patterns and arrive at two observations: (1) the important tokens attended to by different attention heads overlap substantially, and (2) nearby contextual tokens are more critical. Based on these findings, they propose LessIsMore, whose core ideas include merging the tokens selected by different heads into a unified set, enforcing the retention of nearby tokens, and reusing token candidates across most layers.

The method is evaluated on the Qwen series and DeepSeek-R1 models. The results show that, under a 2K token budget, the proposed approach can match or even outperform full attention, while achieving a 1.5× inference speedup.

**Compliance With Llm Reviewing Policy:**

Affirmed.

**Final Justification:**

Author rebuttal well address my concern, thus i recommend for acceptance.

**Key Questions For Authors:**

See Weaknesses.

I'm open to raise the rating if the concerns are mostly addressed.

**Limitations:**

yes

**Strengths And Weaknesses:**

**Strengths:**

* It improves sparse attention from the perspective of stability in reasoning, rather than simply filtering tokens.
* The method is well-motivated and training-free, which is beneficial for deployment.
* The argumentation and experiments are relatively complete.

**Weaknesses:**

* The paper is novel, but the novelty lies more in recombining existing ideas for long-chain reasoning rather than introducing a paradigm shift. The key modules proposed here all have precedents in prior work. For example, TidalDecode introduced persistent cross-layer reuse, while H2O and StreamingLLM also discussed retaining recent tokens.
* The central claim that “token importance in reasoning is a global property, not a head-local one” feels somewhat overstated. Many prior studies on retrieval heads have emphasized head heterogeneity, showing that a small number of heads bear stronger retrieval responsibilities and that CoT is also clearly affected by them. The authors’ observation seems more likely to reflect cross-head overlap under a limited budget, rather than the nonexistence of head-locality. I would recommend softening this claim and expanding the discussion in the related work section accordingly.
* The support for the core observations is not rigorous enough. For the preliminary observational experiments, the main text should present more results across different tasks, layers, and models to fully substantiate the conclusions; some of the material currently placed in the appendix would be better moved into the main paper. In addition, the reported end-to-end speedup is only 1.1× at 16K, suggesting that the method is better suited to extremely long reasoning processes. When the context itself is not very long, the gains are likely to shrink. Moreover, because the selection layer still requires full attention, the approach is not a free lunch; rather, it trades a small number of precise layers for sparsity in most of the remaining layers. This is a reasonable engineering choice, but it also means that the paper’s advantage depends heavily on the reliability of its core observations.
* The experimental coverage is still limited. Datasets such as AIME-24/25 focus mainly on mathematical and scientific reasoning, whereas the paper’s title and overall argument point toward long-horizon reasoning more broadly. To support such a broader claim, additional tasks should be included, such as code generation or multi-turn tool-use reasoning.
* The experimental analysis is not sufficiently thorough. Although the paper demonstrates strong overall performance, it lacks several key analyses, such as: when and why LessIsMore suffers performance drops, and why an approximate attention mechanism can sometimes outperform full attention.

---

> ### Author Rebuttal · Authors · 2026-03-31
>
> We sincerely thank the reviewer for the thoughtful and constructive feedback. We address each concern below.
>
> ---
>
> ### **Summary of Updates**
> - **Novelty:** Clarify that our contribution is identifying a **failure mode of prior sparse attention** and resolving it with **CUSA**. Further validated w. **Quest+CUSA** + gain **50-pt abs. improvement**.
> - **Claim refinement:** *Soften the global vs. head-local statement**.
> - **Evidence:** Clarify broader validation and **speedup scaling behavior**.
> - **Generality:** Add **HumanEval results**.
> - **Analysis:** Explain **performance drops** and why sparse attention can **outperform full attention**.
>
> ---
>
> ### **W1: Novelty**
>
> Our contribution is not any component in isolation, but identifying and resolving a **previously unrecognized failure mode** of sparse attention in long-horizon reasoning (*§2.1*).
>
> Prior methods were designed for short outputs and rely on local token importance (per-head/per-layer/per-step). Our analysis (*Fig.2; §2.2–2.3*) shows that under tight budgets, reasoning depends on a **stable, overlapping subset of globally important tokens across heads**. Per-head selection thus produces locally reasonable but **globally inconsistent token sets**, compounding over long horizons.
>
> To show **CUSA addresses this failure mode**, we apply it to Quest (no hyperparameter change) on AIME-24 with Qwen3-8B for 4 runs, yielding large gains of **50-pt**-abs improvement:
>
> | Method              | 2K | 4K | 6K | 8K |
> |---------------------|---:|---:|---:|---:|
> | Quest + CUSA(Ours)        | 70.0 | **76.7** | **78.2** | **76.7** |
> | Quest               | 18.1 | 46.7 | 49.6 | 59.8 |
> | LessIsMore(Ours)          | **73.7** | 75.8 | 76.7 | **76.7** |
> | Full Attention      | 74.5 | 74.5 | 74.5 | 74.5 |
>
> These results confirm that **CUSA addresses a fundamental failure mode** overlooked by prior sparse attention methods and applies broadly as a principle to all sparse attention mechanisms (beyond our specific system).
>
> ---
>
> ### **W2: Global vs. head-local importance**
> We agree that head heterogeneity is well-established and it was not our intention to dispute it. Instead, our claim is more specific: under tight token budgets in long-chain reasoning, the most critical tokens exhibit **substantial cross-head overlap**, not because heads are homogeneous, but because reasoning tasks impose a shared dependency on a stable global context that transcends individual head specializations. CUSA exploits precisely this overlap. We will soften the claim in §1 and §2.3 to make this distinction explicit, and expand §5 to better situate our finding relative to work on retrieval and induction heads such as [1] and [2].
>
> [1] Liu et al. 2024. RetrievalAttention.
> [2] Xiao et al. 2024. DuoAttention.
>
> ### **W3: Observation support and speedup**
>
> We have provided comprehensive and broader evidence in the appendix, which likely made the core observations feel under-supported in the main text. In revision, we will move representative visualizations of cross-head overlap across reasoning stages from the appendix (currently Fig. 9 and 10 )  into the main body, and more explicitly summarize the broader validation already included across seven models from three model families  (Tab. 10), re-selection layers  (§A.5), and long-context tasks (§A.4).
>
> Re. efficiency: 1.1× e2e speedup at 16K is a system effect rather than a limitation of method. Our kernel is built upon FlashInfer [1], which aggressively optimizes short-context attn (e.g., via split-KV). At 16K tokens, the attn operation represents a relatively small fraction of the end-to-end forward pass latency. However, as generation grows to 32K and 64K—which is increasingly standard for LRMs—attn becomes the dominant bottleneck, and LessIsMore’s e2e speedup scales rapidly to **1.3x and 1.6x**. We clarified this dynamic in §4.3.
>
> [1] Ye et al. 2025. FlashInfer.
>
> ### **W4: Broader experimental coverage**
>
> We add HumanEval (164 questions, 4 runs, Qwen3-8B, gen_len=32K) with same setting. LessIsMore preserves full-attn performance:
>
> | Method        | 2K | 4K | 6K |
> |---------------|---:|---:|---:|
> | LessIsMore    | **89.63** | **88.41** | **89.63** |
> | TidalDecode   | 85.36 | 87.19 | 88.41 |
> | Full Attn| 88.41 | **88.41** | 88.41 |
>
> This shows CUSA and LessIsMore generalized to **code generation**, not just math.
>
> ### **W5: Performance drop and gains over full attn**
>
> Performance of LessIsMore, like any other sparse attention techniques, primarily degrades under **severe budget constraints** where the true volume of logically critical tokens exceeds the fixed sparsity budget.(e.g., **GPQA: 42.80% → 35.47% at 1K** on Deepseek-R1-8B).
>
> Occasional gains over full attention are consistent with prior work (e.g., Quest, TidalDecode) and likely due to:
> 1. **sampling variance** with non-0 temperature and
> 2. **implicit regularization** (removing low-salient and noisy context).
>
> We will add this boundary-case analysis explicitly in the final version.

---

> > ### Author Rebuttal · Reviewer_s3zQ · 2026-04-01
> >
> > I have no more questions.

---

### Official Review · Reviewer_aHqs · 2026-03-13

**Soundness:** 3
**Presentation:** 3
**Significance:** 3
**Originality:** 3
**Overall Recommendation:** 4
**Confidence:** 3

**Summary:**

The authors propose LessIsMore, a training-free sparse attention mechanism designed for long-sequence reasoning models (like DeepSeek-R1 and Qwen3). It aims to solve the memory and computational latency bottlenecks encountered during long-horizon decoding. The study reveals that token importance in reasoning tasks is both global and stable, characterized by: 1) Cross-head spatial locality (critical tokens heavily overlap across different attention heads), and 2) Temporal recency locality (recently generated tokens consistently receive high attention weights).

**Compliance With Llm Reviewing Policy:**

Affirmed.

**Key Questions For Authors:**

Please refer to the weaknesses.

**Limitations:**

Yes

**Strengths And Weaknesses:**

Strengths:

1. LessIsMore uses cross-head aggregation and a fixed recent-window ratio. It is plug-and-play, significantly lowering the barrier to deployment.
2. The results seems good and inspiring.

Weaknesses:
1. The method relies heavily on performing token selection at a specific layer. While the authors validate this choice using a Needle-in-the-Haystack test, is this static layer selection robust against entirely new prompt distributions? Introducing an adaptive layer selection mechanism could make the approach more robust.
2. The current evaluation heavily focuses on math (AIME, MATH500) and complex STEM reasoning (GPQA). Do these cross-head locality patterns hold up in other long-horizon tasks, such as complex coding challenges or long-document logic extraction? Adding boundary tests outside of pure reasoning benchmarks would strengthen the paper's claims of generality.
3. Missing important references:
[1] L1: Controlling How Long A Reasoning Model Thinks With Reinforcement Learning.
[2] Scalable Chain of Thoughts via Elastic Reasoning.

---

> ### Author Rebuttal · Authors · 2026-03-31
>
> We sincerely thank the reviewer for the valuable comments! We address each of reviewer’s concerns below.
>
> ---
>
> ### **Summary of Updates**
> - **Layer robustness:** The re-selection layer is a **stable, model-level property**, and gains arise from **CUSA rather than layer-specific tuning**.
> - **Generality beyond math:** We add **HumanEval** results showing the method extends to **code generation**.
> - **Related work:** We clarify that prior works are **complementary (training-required)** to our **inference-time training-free approach**.
>
> ---
>
> ### **W1: Layer Sensitivity / Robustness**
>
> We thank the reviewer for raising this important clarification.
>
> First, we clarify that the **core contribution of our work is CUSA**, a lightweight and architecture-agnostic component for cross-head unified token selection. The choice of the re-selection layer is inherited from the underlying sparse attention framework (e.g., TidalDecode) and is not the primary focus of our method.
>
> Empirically, this layer choice is not brittle but reflects a stable structural property of reasoning attention. Across all evaluated models, the optimal layer remains consistent within each model family (Table 10), corresponding to depths where semantic reasoning stabilizes.
>
> Importantly, this is a **model-level constant**, not a prompt- or task-dependent parameter. We determine it once per model using a lightweight Needle-in-the-Haystack probe and keep it fixed across all benchmarks. LessIsMore consistently outperforms prior methods across diverse tasks (Figure 5), indicating robustness to distribution shifts.
>
> To further show that the gains arise from CUSA rather than layer-specific tuning, we apply CUSA to Quest and observe substantial improvements (Table W1), with up to **+50-point absolute accuracy gain** at low token budgets.
>
> | Method              | 2K   | 4K   | 6K   | 8K   |
> |---------------------|-----:|-----:|-----:|-----:|
> | Quest               | 18.2 | 46.7 | 49.6 | 59.8 |
> | Quest + CUSA (Ours) | 70.0 | **76.7** | **78.2** | **76.7** |
> | LessIsMore (ours)   | 73.7 | 75.8 | 76.7 | **76.7** |
> | Full Attention      | **74.5** | 74.5 | 74.5 | 74.5 |
>
> **Table W1.** Evaluation results of Quest+CUSA (ours) on AIME-24 using Qwen3-8B (4 runs).
>
> Overall, these results indicate that performance gains stem from **cross-head unified selection**, while the layer choice remains a stable and secondary design inherited from prior frameworks.
>
> ---
>
> ### **W2: Coding Benchmark**
>
> We thank the reviewer for this suggestion. To evaluate generality beyond STEM reasoning, we test on HumanEval [1], a code generation benchmark requiring long-horizon reasoning.
>
> With the same hyperparameter setup (re-selection layer = 12, recency ratio = 0.25), **LessIsMore achieves lossless accuracy at 2K token budget**, while prior methods degrade significantly.
>
> This indicates that the observed locality structure is a **general property of long-horizon reasoning**, including code generation. We will include these results in the final version.
>
> | Method              | 2K     | 4K     | 6K     |
> |---------------------|--------:|--------:|--------:|
> | LessIsMore (Ours)   | **89.63** | **88.41** | **89.63** |
> | TidalDecode         | 85.36 | 87.19 | 88.41 |
> | Full Attention      | 88.41 | **88.41** | 88.41 |
>
> **Table W2.** HumanEval accuracy (164 problems, 4 runs) on Qwen3-8B.
>
> [1] Chen et al., *Evaluating Large Language Models Trained on Code*, arXiv:2107.03374.
>
> ---
>
> ### **W3: Missing References**
>
> We thank the reviewer for suggesting these references. Both L1 [1] and Elastic Reasoning [2] study reasoning efficiency from a **training-time perspective**(e.g., controlling generation length via RL or elastic compute allocation), which is complementary to our **inference-time, training-free sparse attention** approach.
>
> We will include these works in the related work section to better contextualize the broader landscape of efficient reasoning methods.
>
> [1] *Controlling How Long A Reasoning Model Thinks With Reinforcement Learning*
> [2] *Scalable Chain of Thoughts via Elastic Reasoning*

---

> > ### Author Rebuttal · Reviewer_aHqs · 2026-04-03
> >
> > My concerns are resolved.

---

### Decision · Program_Chairs · 2026-04-30

**Decision:**

Accept (regular)

**Comment:**

This paper proposes a new sparse attention mechanism based on the observation that reasoning tokens are global and coherent, where critical tokens are often shared across heads and keep stable during decoding. Therefore, it employs cross-head unified token selection and keeping recent context with a stable recency window reused across different layers. It is training-free and accelerates the inference on long reasoning problems. All reviewers gave a positive score and confirmed the contribution.